# The virulent bacteriophage Henu8 as an antimicrobial synergist against *Escherichia coli*

Fang Zhou,[1] Kexiao Wang,[1] Shuai Ji,[2] Xiaochen Liao,[2] Wenwen Zhang,[3] Tieshan Teng,[1,3] Li Wang,[1] Qiming Li[1,2,3]

**ABSTRACT**    As the overuse of antibiotics has not yet been strictly limited in urban areas, drug-resistant *Escherichia coli* has become a fatal pressure for bacteremia treatment. Considering the outstanding performance of bacteriophages *in vitro*, bacteriophages may serve as an alternative to heal chronic refractory infections. In this study, a 49,890 bp double-stranded circular DNA phage, Henu8, was isolated and was able to lyse the group of *E. coli* strains tested in this study. Prominent biological characterization revealed that the highly adsorbed bacteriophage Henu8 could form a fully transparent plaque with a narrow translucent halo. The optimal multiplicity of infection of the bacteriophage Henu8 was 0.01, with a burst size of 275 PFU/cell. Genomic analysis revealed a G + C content of 44.17% Henu8, in which 65 open reading frames were located, which could be assigned as a new species in the genus *Hanrivervirus* of the subfamily *Tempevirinae*. The effective antibacterial ability and the obvious biofilm destruction and inhibition capability of phage Henu8 were observed. The time-killing assay demonstrated the synergetic potential of Henu8 with antibiotics *in vitro* for *E. coli* eradication. Henu8 has profound medicinal potential in a mouse bacteremia model. These studies indicate that Henu8 is a novel bacteriophage with therapeutic potential alone or in combination with antibiotics for clinical treatment.

**IMPORTANCE** The findings described in this study constitute concrete evidence that it is possible to significantly synergize the antimicrobial activity of bacteriophages and antibiotics. We showed that the newly isolated potent bacteriophage Henu8 lyses *Escherichia coli* rapidly but tends to produce resistant bacteria. The bacteriophage Henu8 has synergistic antimicrobial effects with several antibiotics and is not susceptible to developing resistance. These results provide further evidence that bacterial resistance to phages arises, possibly at an adaptive cost to sensitivity to antibiotics. Therefore, the findings of this study are important for increasing the potential of phages for clinical applications and developing new approaches to improve their therapeutic efficacy against bacterial drug resistance.

**KEYWORDS**    phage therapy, bacteriophage Henu8, *E. coli*, synergistic antibacterial

**Peer Reviewers** Juhee Ahn, Kangwon National University, Chuncheon, Gangwon-do, Republic of Korea; Victor Gonzalez, Universidad Nacional Autonoma de Mexico - Campus Morelos, Cuernavaca, Mexico

Address correspondence to Li Wang, wangli851217@163.com, or Qiming Li, liqiming82@126.com.

Fang Zhou and Kexiao Wang contributed equally to this article. F.Z. was responsible for phage isolation, identification, sequencing, and *in vitro/in vivo* antibacterial assays; K.W. performed statistical analyses, data analysis, and original draft writing.

The authors declare no conflict of interest.

See the funding table on p. 19.

*E*scherichia coli, the predominant aerobic gram-negative species inhabiting the normal intestinal flora and feces of warm-blooded animals and reptiles, has been in close contact with human beings by sharing environments such as soil, water, dust, and sewage, which enables it to be most responsible for nosocomial bacteremia, claiming responsibility for more than 2 million fatalities per year who are killed by both extraintestinal and extraintestinal diseases (1). Among all the hosts that infect *E. coli*, humans, in particular, have a prevalence of more than 90%. Thus, *E. coli* has been well studied by scholars across the globe from the perspectives of epidemiology, evolution, association with humans, and virulence (2, 3). The most acknowledged classification standards list the six well-researched pathotypes of *E. coli* according to their intestinal

behavior and virulence, including Shiga toxin-producing *E. coli*, enteropathogenic *E. coli*, enterotoxigenic *E. coli*, enteroaggregative *E. coli*, diffusely adherent *E. coli*, and enteroinvasive *E. coli* (1). In addition, urinary tract infection induced by uropathogenic *E. coli* has also been considered a severe threat to global health, as have the economic costs associated with its prevalence in the U.S., resulting in $5 billion in economic loss annually (4). Even worse, ever since the declaration of the era of antibiotics, misuse and unregulated prescription authority have exposed civilians on the planet to the grave menace of antibiotic resistance (5–7). Unavoidable barriers lie ahead of those faced by doctors and microbiologists, necessitating more promising and efficient approaches to combat multidrug-resistant (MDR) *E. coli* strains (8). Accordingly, dozens of scholars have considered bacteriophages as effective alternatives to eliminate the inexorable spread of MDR strains (9, 10).

Bacteriophages are viruses that show up where their hosts inhabit, specifically regulating the balance of bacterial distribution in the natural environment or human microbiomes (3, 11). While their attachment to certain receptors of cells provides the foundation for successful infection, this biological behavior determines how narrow the spectrum is, influencing the efficacy and availability of their use (12). With the expansion of antibiotic resistance and the occurrence of MDR bacteria—known by the acronym ESKAPE pathogens (for *Enterococcus faecium*, *Staphylococcus aureus*, *Klebsiella pneumoniae*, *Acinetobacter baumannii*, *Pseudomonas aeruginosa*, and *Enterobacter spp*.)—intensive care units have faced unprecedented tension, as multiple antibiotics have been adopted to resist MDR strain dissemination (7). However, antibiotics cause obvious adverse effects by disrupting the intestinal microbiome, resulting in an awkward dilemma for patients, where a more suitable remedy is imperative. Luckily, owing to their specificity, phages can selectively target and lyse the host bacteria (sometimes a species or strain) while showing little or no adverse effects on the microbiome, which, to a great extent, halts the grave collateral damage caused by antibiotics to bystanders and non-target organisms and decreases the selective pressure induced by antibiotics in MDR bacteria evolution (13). Many bacteriophages targeting clinical isolates have been isolated and shown to exhibit perfect eradication ability (14).

Phage therapy has already been proposed, where preclinical tests in various animals and human tests have been conducted, but limited reporting resources are available (15, 16). During those tests, phage therapy exhibited perfect tolerance and slight adverse effects, where the bacteria hidden in the central nervous system were also eradicated by phages penetrating the blood-brain barrier (9, 17). Considering the narrow spectrum of bacteriophages, normal remedies for phage therapy are predominantly bacteriophage cocktails and bacteriophage bioengineering via mutation of the tail fiber protein (18). In addition to the bacteriophage itself, the proteins encoded by the phages, including the depolymerase, bacteriophage endolysins, and holin, may be of potential interest in confronting infection, which could directly target the specific pathogen or eliminate the bacteria nonspecifically (19–21). All these traits and the efficacy of the phages present unleash the possibility of medicinal use of bacteriophages in MDR bacterial infection treatment.

Although bacteriophages seem to be an ideal alternative for combating the dissemination of MDR *E. coli* strains, phage resistance has been reported in dozens of reports, indicating defects in the sole adoption of bacteriophages (3, 8, 22). To address the acquisition of both phage resistance and antibiotic resistance, a combination of antibiotics and specific phages has been proposed for experimental therapy both *in vitro* and *in vivo* (23–26). Considering that the parameters influencing the combined effects are unpredictable, the probable outcomes may be classified as follows: (i) additive effect, which refers to the combined effect of both phage and antibiotics; (ii) synergetic effect, which indicates that the total efficacy is greater than that of each individual combination; and (iii) no effect, in which no greater efficacy was detected than when it was used individually (12). In that case, antibiotic selection, phage titer, sequence of adoption, and doses should be well considered or, to some extent, personalized (27, 28). Underlying the

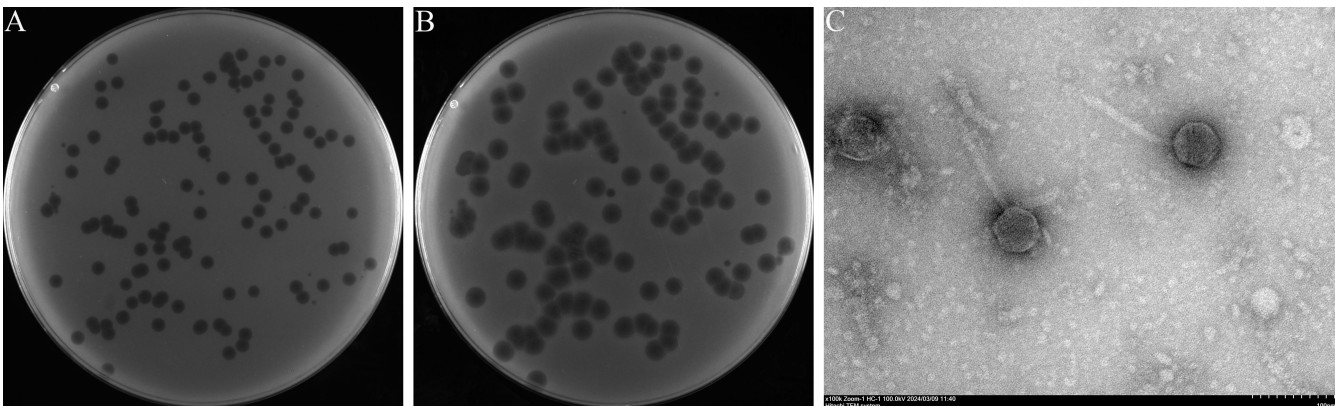

**FIG 1** Morphology of the bacteriophage Henu8. Phage plaques formed on a lawn of *E. coli* BW25113 incubated at 37°C for 12 h (A) and 24 h (B). (C) Transmission electron micrograph of the bacteriophage Henu8.

synergetic effect may be the molecular interaction, which has been confirmed by several researchers. The synergetic effects of phages have been revealed in dozens of drugs for which various therapies have been adopted (12, 29). In summary, the combination still holds promising medicinal potential in severe infections induced by MDR strains because of its low resistance frequency and ability to regain susceptibility (23, 30, 31).

Bacteriophages of *E. coli* can be easily isolated with great ease from their prevalence in natural environments, including sewage, soil, and contaminated food. The isolation of novel bacteriophages could further our understanding of the biological behavior and therapeutic use of bacteriophages in infection treatment. In this study, by using *E. coli* BW25113 as a host, we successfully isolated and purified the *E. coli* bacteriophage Henu8 and conducted a series of experiments to verify its genomic characteristics and biological characteristics. Furthermore, the bacteriophage Henu8 has shown synergetic potential with several antibiotics and has the potential to be used in combination with other clinical infections. Finally, by establishing a bacteremia model, we verified that the bacteriophage Henu8 was capable of eliminating *E. coli in vivo*, laying the foundation for its use clinically against *E. coli* infection.

## RESULTS

### Isolation and morphology of the bacteriophage Henu8

The bacteriophage Henu8 was isolated from sewage by using *E. coli* BW25113 as the host strain. A fully transparent and regular plaque with a narrow halo ring was observed after plating on the double-layer agar after incubation at 37°C for 12 h (Fig. 1A). As the incubation time increased, the plaque gradually enlarged, and the surrounding halo became more pronounced (Fig. 1B). However, it is worth noting that after five rounds of purification by picking a single plaque and inoculating it with a bacterial mixture, large plaques and small plaques still coexisted in one plate (Fig. 1A and B). Transmission electron microscopy was subsequently used to observe the morphology of bacteriophage Henu8. The bacteriophage Henu8 had a regular icosahedral head with an average diameter of 55.2 nm and a long tail with an average length of 117 nm (Fig. 1C).

### Host range analysis

To evaluate the host spectrum of bacteriophage Henu8 infection, a series of bacterial strains, including *E. coli*, *K. pneumoniae*, *P. aeruginosa*, and *A. baumannii*, were tested. The results demonstrated that the bacteriophage Henu8 exhibits a narrow host range, specifically infecting only a limited number of *E. coli* strains. Notably, it showed no infectivity against *K. pneumoniae*, *P. aeruginosa*, or *A. baumannii* (Table 1). Astonishingly, A57, an MDR *E. coli* strain isolated from the First Affiliated Hospital of Henan University

**TABLE 1** Strains used for bacteriophage Henu8 host range analysis[a]

| Species | Strain | Susceptibility | Origin |
|---|---|---|---|
| *E. coli* | BW25113 | ++++ | Shanghai Microbial Collection Center |
| | BL21 (DE3) | ++++ | TAKARA |
| | DH5α | ++++ | TAKARA |
| | A57 | +++ | The First Affiliated Hospital of Henan University |
| | A30 | – | The First Affiliated Hospital of Henan University |
| | A28 | – | The First Affiliated Hospital of Henan University |
| | A25 | – | The First Affiliated Hospital of Henan University |
| *A. baumannii* | A.ba-1 | – | The First Affiliated Hospital of Henan University |
| | A.ba-2 | – | The First Affiliated Hospital of Henan University |
| *P. aeruginosa* | P.ae-1 | – | The First Affiliated Hospital of Henan University |
| | P.ae-2 | – | The First Affiliated Hospital of Henan University |
| *K. pneumoniae* | ATCC700603 | – | Shanghai Microbial Collection Center |
| | K.p-2 | – | The First Affiliated Hospital of Henan University |

[a]++++, plaques at $10^6$; +++, plaques at $10^5$; ++, plaques at $10^4$; –, no plaques.

(Table S1), could also be effectively lysed by the bacteriophage Henu8. The results suggest that bacteriophage Henu8 has potential application for *E. coli* infection.

## Biological characterization of the bacteriophage Henu8

To determine the optimal multiplicity of infection (MOI) of the bacteriophage Henu8 for *E. coli* BW25113, the bacteriophage Henu8 was mixed with *E. coli* BW25113 ($10^8$ CFU/mL) at different MOIs. The phage titer reached a maximum at an MOI of 0.01, suggesting that the optimal MOI of the bacteriophage Henu8 was 0.01 (Fig. 2A). Adsorption tests revealed that 90% of the phage particles could successfully adsorb to the cells after 15 min of incubation (Fig. 2B). The one-step growth curve at the optimal MOI of 0.01 suggested that bacteriophage Henu8 had a relatively short latent period (less than 5 min) and could rapidly reach the plateau period in 35 min with the burst size of 275 PFU/Cell (Fig. 2C).

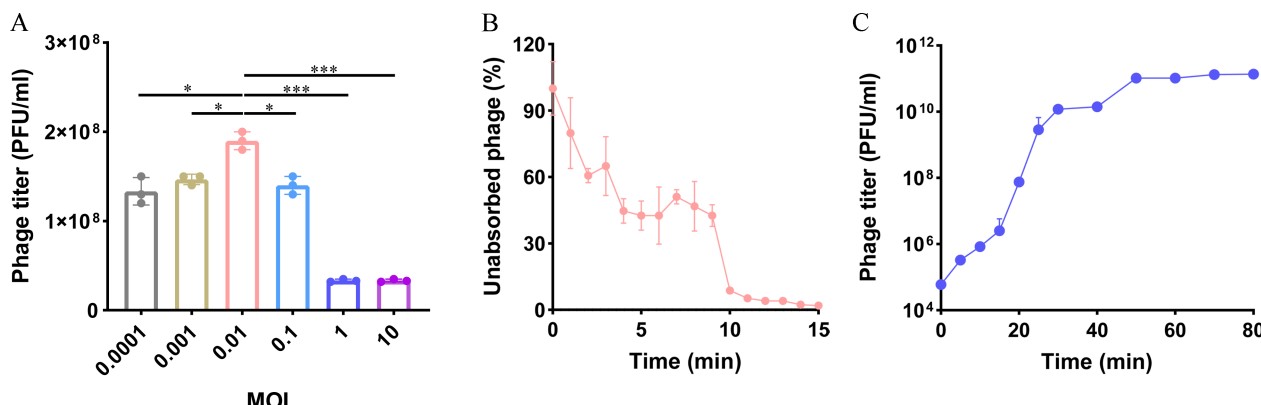

**FIG 2** Biological characterization of the bacteriophage Henu8. (A) Determination of the optimal MOI. The bacteriophage Henu8 was mixed with an *E. coli* BW25113 suspension at the different MOIs, and after 2 h of incubation with shaking at 37°C, the optimal MOI was determined via the plaque counting method. (B) Adsorption curve of the bacteriophage Henu8. The bacteriophage Henu8 was mixed with *E. coli* BW25113 to reach the optimal MOI of 0.01. While the mixture was incubated with shaking, samples were taken every minute. (C) One-step growth curve of the bacteriophage Henu8. The bacteriophage Henu8 was mixed with *E. coli* BW25113 at an MOI of 0.01. After 15 min of adsorption, the suspension was incubated with shaking for 80 min, after which samples were collected and diluted for plaque counting at the indicated intervals. The data are presented as the means ± SD. Statistical analysis was performed via one-way analysis of variance following Dunnett's multiple comparisons test. *$P < 0.05$, **$P < 0.01$, and ****$P < 0.0001$.

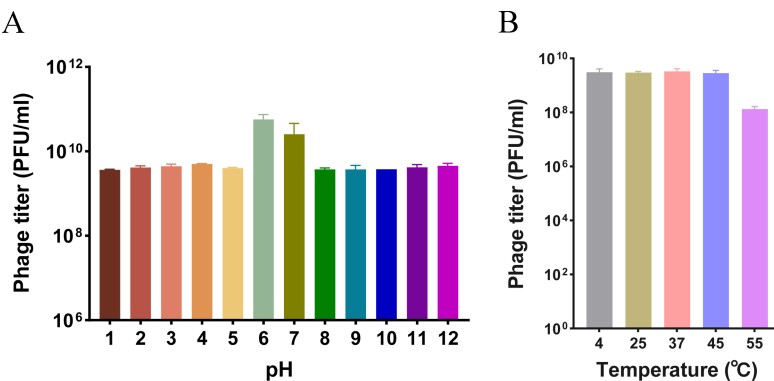

**FIG 3** Sensitivity of the bacteriophage Henu8 to pH and temperature. (A) pH stability of the bacteriophage Henu8. The bacteriophage Henu8 was mixed in saline at different pH values (1–12), and samples were collected for PFU counting after incubation with shaking for 60 min. (B) Thermal stability of bacteriophage Henu8. The purified bacteriophage Henu8 suspensions were incubated at 4°C, 25°C, 37°C, 45°C, and 55°C for 12 h, after which the phage titer was detected.

## Sensitivity of the bacteriophage Henu8 to pH and temperature

For the storage of bacteriophages and instruction in further *in vivo* experiments, revealing the basic characterization of bacteriophage Henu8 is crucial. Therefore, to characterize the stability and sensitivity of the bacteriophage Henu8 under various conditions, several sensitivity tests were conducted, the results of which were quantified via the phage plaque counting method (32). The bacteriophage Henu8 exhibited steady activity under different pH values, even under extreme conditions (pH = 1 or pH = 12), where bacteriophage Henu8 still presented a relatively high infection ability (90% decrease compared with that at pH = 7; Fig. 3A). Additionally, we tested bacteriophage Henu8 infection in the host *E. coli* BW25113 at different temperatures. The bacteriophage Henu8 maintained stable titers when incubated at temperatures ranging from 4°C to 45°C for 12 h, while a significant reduction in phage titer was observed at 55°C (Fig. 3B). Moreover, the bacteriophage Henu8 could be easily affected by UV destruction (30W) in less than 30 min (Fig. S1A). Organic solvents, including chloroform and diethyl ether, have also been used to analyze phage plaque formation by the bacteriophage Henu8. Chloroform significantly affected the viability of bacteriophage Henu8 to some extent (99%), whereas ether caused a 10% decrease in activity (Fig. S1B).

## Genomic characterization of the bacteriophage Henu8

The genome of the bacteriophage Henu8 is a double-stranded circular DNA with a total length of 49,890 bp, harboring a G + C content of 44.17%. According to what Softberry presented, a total of 65 open reading frames (ORFs) were encoded by the bacteriophage Henu8 genome, 37 of which were annotated as functional proteins via BLASTp (Fig. 4A; Table 2). In addition, no drug resistance genes or virulence factors were detected in the genome of the bacteriophage Henu8 (Fig. 4A). To verify the circular configuration of the bacteriophage Henu8 genome, we conducted agarose gel electrophoresis on the enzymatic digestion products generated by the two unique restriction endonucleases, *EcoR* I and *Xba* I, within its genome (Fig. S2). The bacteriophage Henu8 genome was digested into two distinct fragments when codigested with *EcoR* I and *Xba* I, while digestion with either *EcoR* I or *Xba* I alone yielded a single fragment (Fig. 4B). The above results indicated that the genome of the bacteriophage Henu8 is indeed double-strand circular DNA and lay the foundation for future engineering of the bacteriophage Henu8 as an antibacterial therapy.

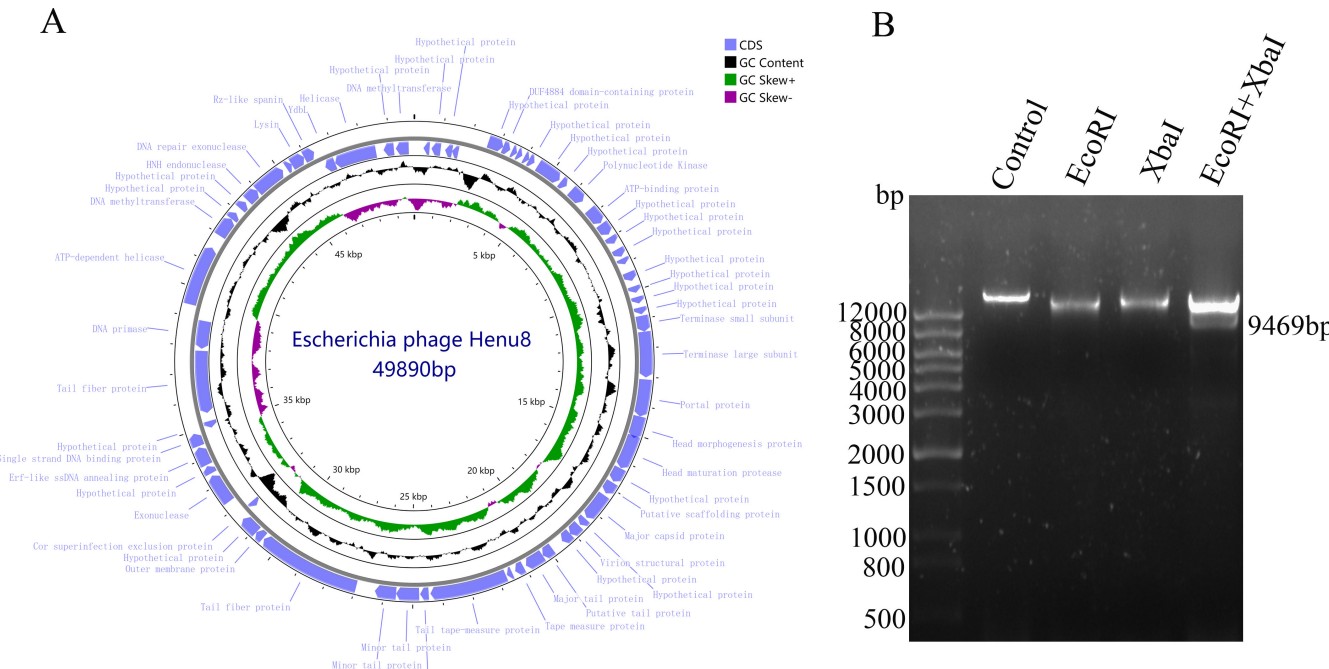

**FIG 4** Genome map of the bacteriophage Henu8. (A) All 65 protein-encoding ORFs were annotated in the genome map, where the individual length of the ORFs and the full length of the bacteriophage Henu8 were annotated. (B) Digestion of bacteriophage Henu8 genome via *EcoR* I and *Xba* I. *EcoR* I and *Xba* I were adopted to specifically cleave the genome of the bacteriophage Henu8, where two separated fragments were visualized via agarose gel by electrophoresis.

## Evolutionary analysis of bacteriophage Henu8

To illustrate the evolutionary relationships of the bacteriophage Henu8 with other phages, a phylogenetic tree was constructed on the basis of the conserved phage major capsid protein (ORF 30), putative portal protein (ORF 25), and terminase large subunit (ORF 24). According to major capsid protein analysis, the bacteriophage Henu8 was highly correlated with *Shigella* phage pSf-1 and *Escherichia* phage vB EcoS G29-2 (Fig. 5A). Like the putative portal protein presented, the bacteriophage Henu8 was highly related to the *Escherichia* phage vB EcoS G29-2 (Fig. 5B). The results of the terminase large subunit analysis revealed that the bacteriophage Henu8 was relatively highly correlated with *Shigella* phage pSf-1 and *Escherichia* phage vB EcoS G29-2 (Fig. 5C). In summary, the bacteriophage Henu8 shares a close evolutionary relationship with *Escherichia* phage vB EcoS G29-2 and *Shigella* phage pSf-1.

## Evaluation of the bacteriophage Henu8 against *E. coli in vitro*

To determine whether the bacteriophage Henu8 could be applied in phage therapy for *E. coli* infection, its lytic ability was characterized. By coincubating the bacteriophage Henu8 and *E. coli* BW25113 at different MOIs (10, 1, 0.1, 0.01, 0.001, and 0.0001), where bacteria in the logarithmic phase without phage infection served as a control, the bacteriophage Henu8 exhibited excellent inhibition of the *E. coli* strain and could effectively halt the growth of *E. coli* BW25113 within 4 h (Fig. 6A). The anti-MDR *E. coli* ability of the bacteriophage Henu8 was also assessed by the MDR strain *E. coli* A57, providing concrete evidence that the bacteriophage Henu8 indeed has anti-MDR *E. coli* ability, as verified by the change in the $OD_{600}$ (Fig. S3). However, with the possibility of bacteria escaping the infection or the emergence of phage resistance, the $OD_{600}$ gradually increased after 4 h of coincubation (Fig. 6A). Biofilms are composed of a matrix of bacteria in the extracellular environment, leading to continuous infection, which is difficult to eradicate. Biofilm destruction and inhibition ability are usually considered the parameters determining clinical use in combating infection (33). Therefore, we tested

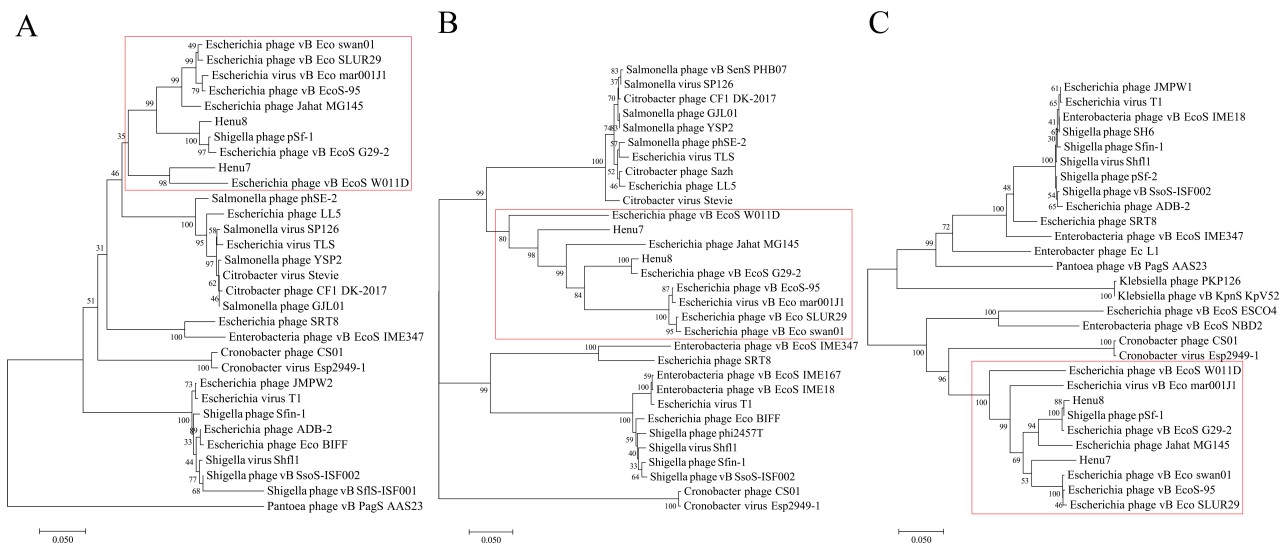

**FIG 5** Phylogenetic analysis of the bacteriophage Henu8. The phylogenetic tree was constructed on the basis of the conserved conservative amino acid sequences of the major capsid protein (A), putative portal protein (B), and terminase large subunit (C) by using MEGA 7.

the biofilm inhibition capability first by mixing phages at different concentrations with mature biofilms. After coincubation at 37°C overnight and crystal violet (CV) staining, a significant difference ($P < 0.0001$) in the absorbance of the attached biomass was detected between the control group and the groups treated with different MOIs (10, 1, 0.1, 0.01, 0.001, and 0.0001), suggesting the obvious ability of the bacteriophage Henu8 to inhibit biofilm formation (Fig. 6B). Additionally, we tested the disruptive effect of bacteriophage henu8 on mature biofilms. The results indicated that the mature biofilms were significantly destroyed when the bacteriophage Henu8 was incubated with *E. coli* BW25113 ($P < 0.0001$; Fig. 6C). These results revealed that the bacteriophage Henu8 has significant lytic, biofilm inhibition, and biofilm destruction capabilities.

## The bacteriophage Henu8 and antibiotics have synergistic antibacterial effects *in vitro*

Although antibiotics have been universally adopted in infection treatment plans, the dissemination of drug resistance has left traditional medicinal therapy in no place. Therefore, for MDR bacterial treatment, bacteriophage therapy seems to be an option because of its distinct specificity and low frequency of drug resistance. However, the use of phages in infection treatment is usually accompanied by phage resistance (34). Considering the acquisition of both phage resistance and drug resistance, the combination of antibiotics and phages seems to be a unique strategy because it decreases both the probability of phage resistance and the probability of antibiotic resistance by reducing the number of doses of antibiotics used, alleviating the selective pressure that high-dose antibiotics bring (30). In this study, we tested a few dozen antibiotics to determine the synergetic potential of the bacteriophage Henu8. Bacteriophage Henu8 at an MOI of 0.01 exhibited prominent synergetic capability with ceftriaxone, moxifloxacin, kanamycin, tetracycline, and polymyxin E at $1/2 \times$ MIC, whereas no visible antibiotic efficacy was observed at $1/2 \times$ MIC for antibiotics alone against *E. coli* at $10^8$ CFU/mL via plate counting methods (Fig. 7A through E). In contrast, when combined with meropenem, ampicillin, and rifampicin, the effects of the combination were equivalent to those of the bacteriophage Henu8 alone, suggesting that no synergetic potential was observed with these antibiotics (Fig. 7F through H).

To determine the probable interaction of antibiotics with the bacteriophage Henu8, an *E. coli* suspension with $1/2 \times$ MIC antibiotics was mixed with the bacteriophage Henu8 at an MOI of 0.01. After incubation for 4 min, samples were taken and centrifuged

**TABLE 2** Features of the open reading frames (ORFs) of bacteriophage Henu8

| ORF | Strand[a] | Start | Stop | Length (AA) | Predicted protein function | Best-match BLASTp result | Query cover | E-value | Identity | Accession | MW (kDa) |
|---|---|---|---|---|---|---|---|---|---|---|---|
| 1 | – | 339 | 566 | 75 | Hypothetical protein | Escherichia phage aaroes | 100% | 3e-46 | 98.67% | YP_009901884.1 | 8.5 |
| 2 | – | 638 | 979 | 113 | Hypothetical protein | Escherichia phage PGN590 | 100% | 9e-73 | 94.69% | YP_009902188.1 | 13.1 |
| 3 | – | 1,148 | 1,414 | 88 | Hypothetical protein | Shigella phage SFP17 | 100% | 4e-49 | 85.23% | QHB43452.1 | 9.6 |
| 4 | – | 1,407 | 1,646 | 79 | Hypothetical protein | Escherichia phage vojen | 100% | 6e-49 | 97.47% | YP_009901717.1 | 8.9 |
| 5 | + | 2,591 | 3,121 | 176 | Hypothetical protein | Shigella phage SFP17 | 88% | 4e-107 | 97.44% | QHB43448.1 | 19.5 |
| 6 | + | 3,121 | 3,381 | 86 | DUF4884 domain-containing protein | Escherichia phage JLBYU41 | 100% | 2e-53 | 95.35% | UGO55899.1 | 9.3 |
| 7 | + | 3,454 | 3,639 | 61 | Hypothetical protein | Escherichia phage vB_EcoS-95 | 100% | 3e-35 | 100% | YP_009818064.1 | |
| 8 | + | 3,655 | 3,840 | 61 | Hypothetical protein | Shigella phage pSf-1 | 100% | 2e-34 | 100% | YP_008059759.1 | 7 |
| 9 | + | 3,926 | 4,111 | 61 | Hypothetical protein | Salmonella phage vB_SenS_Sasha | 96% | 2e-24 | 83.05% | YP_009787778.1 | 6.9 |
| 10 | + | 4,111 | 4,320 | 69 | Hypothetical protein | Salmonella phage vB_SenS_Sasha | 98% | 4e-27 | 70.59% | YP_009787777.1 | 7.8 |
| 11 | + | 4,475 | 5,362 | 295 | Hypothetical protein | Escherichia phage egaa | 100% | 0 | 91.86% | YP_009904901.1 | 33.5 |
| 12 | + | 5,436 | 5,681 | 81 | Hypothetical protein | Escherichia phage vB_EcoD_Poky | 100% | 5e-42 | 82.72% | UGO52309.1 | 9.2 |
| 13 | + | 5,901 | 6,440 | 179 | Polynucleotide Kinase | Escherichia phage vojen | 100% | 2e-128 | 98.32% | YP_009901726.1 | 20.4 |
| 14 | + | 6,803 | 7,408 | 201 | ATP-binding protein | Escherichia phage JLBYU09 | 100% | 5e-145 | 98.01% | UGL62473.1 | 23.4 |
| 15 | + | 7,447 | 7,881 | 144 | Hypothetical protein | Escherichia phage BEK7 | 92% | 1e-89 | 96.99% | QGH76997.1 | 16.9 |
| 16 | + | 7,981 | 8,244 | 87 | Hypothetical protein | Escherichia phage grams | 100% | 1e-57 | 100% | YP_009901814.1 | 10.7 |
| 17 | + | 8,464 | 8,751 | 95 | Hypothetical protein | Escherichia phage vB_EcoD_Fulano1 | 100% | 3e-64 | 98.95% | UGV22683.1 | 11.3 |
| 18 | + | 8,827 | 9,039 | 70 | Hypothetical protein | Shigella phage pSf-1 | 100% | 1e-40 | 100% | YP_008059775.1 | 7.9 |
| 19 | + | 9,341 | 9,655 | 104 | Hypothetical protein | Escherichia phage KarlJaspers | 100% | 4e-68 | 95.19% | QXV82337.1 | 12 |
| 20 | + | 9,854 | 10,084 | 76 | Hypothetical protein | Salmonella phage slyngel | 100% | 9e-50 | 100% | YP_009901471.1 | 8.7 |
| 21 | + | 10,232 | 10,465 | 77 | Hypothetical protein | Escherichia phage grams | 100% | 2e-47 | 97.4% | YP_009901821.1 | 8.8 |
| 22 | + | 10,604 | 10,843 | 79 | Hypothetical protein | Shigella phage pSf-1 | 100% | 2e-48 | 100% | YP_008059781.1 | 8.8 |
| 23 | + | 10,894 | 11,445 | 183 | Terminase small subunit | Escherichia phage JLBYU41 | 100% | 7e-132 | 99.45% | UGO55921.1 | 20.6 |
| 24 | + | 11,445 | 13,013 | 522 | Terminase large subunit | Escherichia phage vojen | 100% | 0 | 99.62% | YP_009901742.1 | 59.2 |
| 25 | + | 13,082 | 14,362 | 426 | Portal protein | Shigella phage ESh4 | 100% | 0 | 99.53% | URY10802.1 | 47.8 |
| 26 | + | 14,362 | 15,126 | 254 | Head morphogenesis protein | Escherichia phage vB_EcoD_Fulano1 | 100% | 0 | 99.21% | UGV22606.1 | 29.2 |
| 27 | + | 15,047 | 16,243 | 398 | Head maturation protease | Escherichia phage herni | 92% | 0 | 99.19% | YP_009901689.1 | 43.3 |
| 28 | + | 16,256 | 16,768 | 170 | Hypothetical protein | Salmonella phage slyngel | 100% | 4e-119 | 98.82% | YP_009901479.1 | 17.6 |
| 29 | + | 16,814 | 17,248 | 144 | Putative scaffolding protein | Escherichia phage JLBYU09 | 100% | 6e-94 | 96.53% | UGL62492.1 | 15.3 |
| 30 | + | 17,332 | 18,306 | 324 | Major capsid protein | Escherichia phage vB_EcoD_Fulano1 | 100% | 0 | 99.38% | UGV22610.1 | 36.2 |
| 31 | + | 18,363 | 18,635 | 90 | Hypothetical protein | Shigella phage pSf-1 | 100% | 8e-57 | 100% | YP_008059791.1 | 10 |
| 32 | + | 18,683 | 19,099 | 138 | Virion structural protein | Escherichia phage haarsle | 100% | 1e-97 | 100% | YP_009902057.1 | 15.7 |
| 33 | + | 19,096 | 19,467 | 123 | Hypothetical protein | Escherichia phage EC147 | 100% | 5e-82 | 99.19% | URC25719.1 | 13.9 |
| 34 | + | 19,893 | 20,291 | 132 | Putative tail protein | Escherichia phage vB_EcoD_Fulano1 | 100% | 5e-91 | 98.48% | UGV22615.1 | 15.4 |
| 35 | + | 20,297 | 20,959 | 220 | Major tail protein | Escherichia phage JakobBernoulli | 99% | 2e-161 | 99.09% | QXV80689.1 | 24.1 |
| 36 | + | 21,059 | 21,376 | 105 | Tape measure protein | Shigella phage pSf-1 | 100% | 1e-72 | 100% | YP_008059797.1 | 12 |
| 37 | + | 21,496 | 21,687 | 63 | Tape measure protein | Escherichia phage vB_EcoS_G29-2 | 100% | 1e-39 | 100% | YP_009901586.1 | 7.5 |
| 38 | + | 21,727 | 24,399 | 890 | Tail tape-measure protein | Shigella phage pSf-1 | 100% | 0 | 99.55% | YP_008059799.1 | 97.2 |

*(Continued on next page)*

TABLE 2 Features of the open reading frames (ORFs) of bacteriophage Henu8 (*Continued*)

| ORF | Strand[a] | Start | Stop | Length (AA) | Predicted protein function | Best-match BLASTp result | Query cover | E-value | Identity | Accession | MW (kDa) |
|---|---|---|---|---|---|---|---|---|---|---|---|
| 39 | + | 24,456 | 24,752 | 98 | Minor tail protein | *Escherichia* phage vB_EcoS_G29-2 | 100% | 1e-65 | 100% | YP_009901588.1 | 11 |
| 40 | + | 24,784 | 25,575 | 263 | Minor tail protein | *Escherichia* phage JLBYU41 | 100% | 0 | 99.62% | UGO55938.1 | 29.2 |
| 41 | + | 25,575 | 26,306 | 243 | Minor tail protein | *Escherichia* phage vojen | 100% | 0 | 99.18% | YP_009901759.1 | 28 |
| 42 | + | 26,964 | 30,542 | 1,192 | Tail fiber protein | *Escherichia* phage grams | 100% | 0 | 99.08% | YP_009901843.1 | 132.1 |
| 43 | + | 30,586 | 30,891 | 101 | Outer membrane protein | *Escherichia* phage grams | 100% | 1e-63 | 99.01% | YP_009901844.1 | 10.6 |
| 44 | + | 30,893 | 31,534 | 213 | Hypothetical protein | *Shigella* phage SFP17 | 100% | 7e-150 | 98.12% | QHB43400.1 | 22.5 |
| 45 | – | 31,565 | 31,792 | 75 | Cor superinfection exclusion protein | *Escherichia* phage JLBYU09 | 100% | 2e-47 | 97.33% | UGL62510.1 | 8.3 |
| 46 | + | 32,295 | 33,356 | 353 | Exonuclease | *Escherichia* phage EC147 | 100% | 0 | 99.72% | URC25735.1 | 40.5 |
| 47 | + | 33,438 | 33,716 | 92 | Hypothetical protein | *Escherichia* phage vB_EcoD_Fulano1 | 100% | 2e-58 | 98.91% | UGV22630.1 | 10.4 |
| 48 | + | 33,762 | 34,418 | 218 | Erf-like ssDNA annealing protein | *Escherichia* phage PGN590 | 100% | 1e-158 | 99.54% | YP_009902206.1 | 24.6 |
| 49 | + | 34,463 | 34,915 | 150 | Single strand DNA binding protein | *Escherichia* phage damhaus | 100% | 8e-107 | 100% | YP_009901971.1 | 17.4 |
| 50 | – | 34,930 | 35,187 | 85 | Hypothetical protein | *Shigella* phage pSf-1 | 100% | 8e-53 | 100% | YP_008059811.1 | 9 |
| 51 | – | 35,499 | 37,820 | 773 | Tail fiber protein | *Escherichia* phage JLBYU41 | 99% | 0 | 61.2% | UGO55972.1 | 83 |
| 52 | – | 37,898 | 38,932 | 344 | DNA primase | *Escherichia* phage vojen | 100% | 0 | 97.97% | YP_009901774.1 | 39.2 |
| 53 | + | 39,401 | 41,488 | 695 | ATP-dependent helicase | *Escherichia* phage JLBYU09 | 100% | 0 | 98.42% | UGL62523.1 | 79.6 |
| 54 | + | 41,969 | 42,670 | 233 | DNA methyltransferase | *Escherichia* phage morffagbaw | 100% | 7e-173 | 98.71% | CAI9420983.1 | 26.7 |
| 55 | + | 42,675 | 42,932 | 85 | Hypothetical protein | *Escherichia* phage vB_EcoD_Fulano1 | 100% | 2e-53 | 95.29% | UGV22642.1 | 10.3 |
| 56 | + | 43,135 | 43,419 | 94 | Hypothetical protein | *Shigella* phage ESh4 | 100% | 9e-61 | 97.87% | URY10758.1 | 11.1 |
| 57 | + | 43,510 | 43,983 | 157 | His-Asn-His (HNH) endonuclease | *Escherichia* phage P818 | 96% | 5e-52 | 53.9% | UOX38524.1 | 18.4 |
| 58 | + | 43,989 | 45,101 | 370 | DNA repair exonuclease | *Escherichia* phage vojen | 99% | 0 | 99.19% | YP_009901782.1 | 41.8 |
| 59 | + | 45,248 | 45,463 | 71 | Holin | *Escherichia* phage vB_EcoS_G29-2 | 100% | 3e-42 | 100% | YP_009901614.1 | 7.5 |
| 60 | + | 45,463 | 45,951 | 162 | Lysin | *Escherichia* phage EC147 | 100% | 1e-116 | 99.38% | URC25754.1 | 18.2 |
| 61 | + | 45,954 | 46,340 | 128 | Rz-like spanin | *Escherichia* phage PGN590 | 100% | 2e-83 | 98.44% | YP_009902195.1 | 13.9 |
| 62 | – | 46,473 | 46,865 | 130 | YdbL | *Escherichia* phage aaroes | 100% | 9e-92 | 100% | YP_009901878.1 | 15.4 |
| 63 | – | 46,868 | 48,451 | 527 | Helicase | *Escherichia* phage grams | 100% | 0 | 100% | YP_009901870.1 | 58.3 |
| 64 | – | 48,739 | 49,125 | 128 | Hypothetical protein | *Salmonella* phage slyngel | 100% | 7e-89 | 99.22% | YP_009901523.1 | 14.8 |
| 65 | – | 49,197 | 49,679 | 160 | DNA methyltransferase | *Escherichia* phage EC147 | 100% | 3e-110 | 95.62% | URC25762.1 | 18.1 |

[a]Strand symbols: +, sense/coding strand; –, antisense/template strand.

to collect the supernatants, and the adsorption rate was determined via the soft agar overlay method. The results showed that no significant differences were observed for the antibiotics except for ceftriaxone sodium (Fig. 8I), suggesting that the ability of the bacteriophage Henu8 to adsorb to bacteria was barely affected by antibiotics.

## Evaluation of the bacteriophage Henu8 against *E. coli* BW25113 *in vivo*

Given the ability of bacteriophage Henu8 to eradicate bacteria *in vitro* and its synergetic potential, we tested the *in vivo* antibacterial activity of the bacteriophage Henu8 to verify its therapeutic potential. To illustrate the antibacterial ability of the bacteria, the C57BL/6N mice (*n* = 5) were injected with $5 \times 10^9$ CFU/mL *E. coli* BW25113 to establish an infection model. After 1 h, different concentrations of the bacteriophage Henu8 (MOI of 10, 1, 0.1, or 0.01) were suspended intraperitoneally for treatment, and saline injection served as a negative control. The mice without phage therapy died of bacteremia within 3 days, whereas phage interference increased the survival of the mice with bacteremia to varying extents (Fig. 8A). The group with an MOI of 0.01 had survival rates that were 40%

and 20% lower than those of the groups with MOIs of 0.1 and 1. Among all the groups, the injection of the bacteriophage Henu8 at an MOI of 10 had the most prominent curative effect, increasing the survival rate to 80% compared with that of the negative control. The *E. coli* A57 strain was also included in our assessment, where the same doses of bacteria and the bacteriophage Henu8 were injected. The high virulence of A57 was verified by the total death of the mice within 1 day, and the therapeutic effect was confirmed by improvements in survival expectations (Fig. S4). Next, the bacterial load in various organs was analyzed by injecting bacteria ($5 \times 10^8$ CFU/mL) into the C57BL/6N mice ($n = 3$), and the bacteriophage Henu8 at different concentrations (MOI = 0.01, 0.1, 1, or 10) was injected for treatment. After 48 h, the mice were dissected, and CFU counting of living bacteria was conducted to detect the bacterial loads in various organs, including the heart, liver, kidney, lung, and spleen. All the organs of the control group presented the greatest bacterial loads, while the bacterial loads of the treatment groups decreased sequentially as the MOIs increased (Fig. 8B). Even at an MOI of 0.01, bacteriophage Henu8 had prominent antibacterial activity in the liver, spleen, lung, and kidney, but not in the heart (Fig. 8B). These findings indicated that the bacteriophage Henu8 has prominent clinical therapeutic potential. To further validate the effect of bacteriophage Henu8 in combination with antibiotics, we utilized bacteriophage Henu8 in combination with moxifloxacin to treat bacteremic mice. The results revealed that the bacteriophage Henu8 or moxifloxacin alone increased the survival rate of the mice by 40%–50%, while the survival rate of the mice treated with the combination treatment reached 80% (Fig. 8C).

## DISCUSSION

Bacteriophages are abundant resources in the environment, leveraging host bacteria as replication factories to achieve self-replication and balance the distribution and growth of bacteria in certain microenvironments (35, 36). As the predators of bacteria, bacteriophages can be divided into mainly two main types according to their propagation properties: lysogenic bacteriophages and lytic bacteriophages. Compared with lysogenic phages, which are designed to integrate their genome into host genome fragments to attain eternal coexistence and confer novel host traits, phages with lytic propagation are prone to hunt for and lyse the host bacteria for self-reproduction and release their progeny viruses, which extends our vision in treating refractory infections by adopting the lytic ones to concoct "cocktails," mixtures of various phages with intense lytic capability, and to confront MDR infections (37). Therefore, the greater our understanding of phages, the greater the number of applications we are about to have in infection resistance. In this study, we used *E. coli* BW25113 as the host and successfully isolated

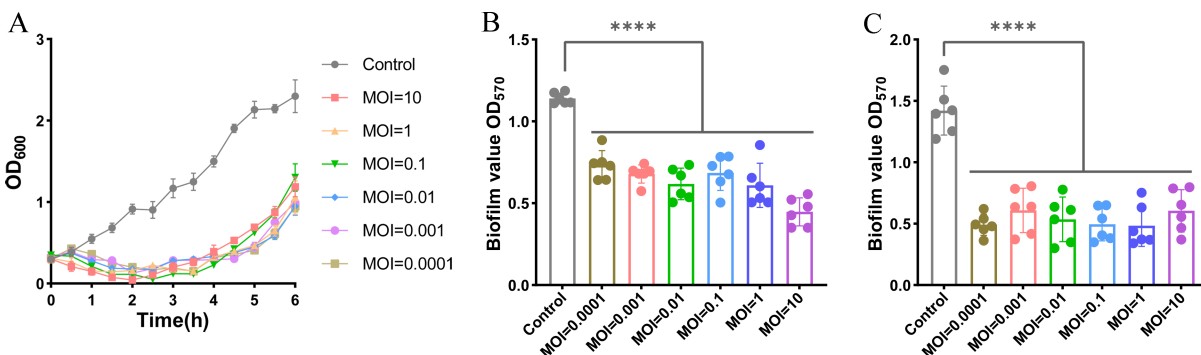

**FIG 6** *In vitro* evaluation of lytic efficacy against *E. coli* BW25113. (A) Evaluation of the inhibition of the bacteriophage Henu8. The bacteriophage Henu8 was coincubated with *E. coli* BW25113 at the indicated MOIs in a shaking incubator, where samples were taken every 30 min for absorbance tests at 600 nm. (B) Biofilm inhibition of the bacteriophage Henu8. The bacterial mixture was coincubated with the bacteriophage Henu8 to inhibit the growth of the biofilm. (C) Biofilm destruction test of the bacteriophage Henu8. Preformed biofilms of bacteria were treated with bacteriophage Henu8 after 24 h. The inhibition and destruction abilities were assessed by measuring the decrease in absorbance at 570 nm. ****$P < 0.0001$.

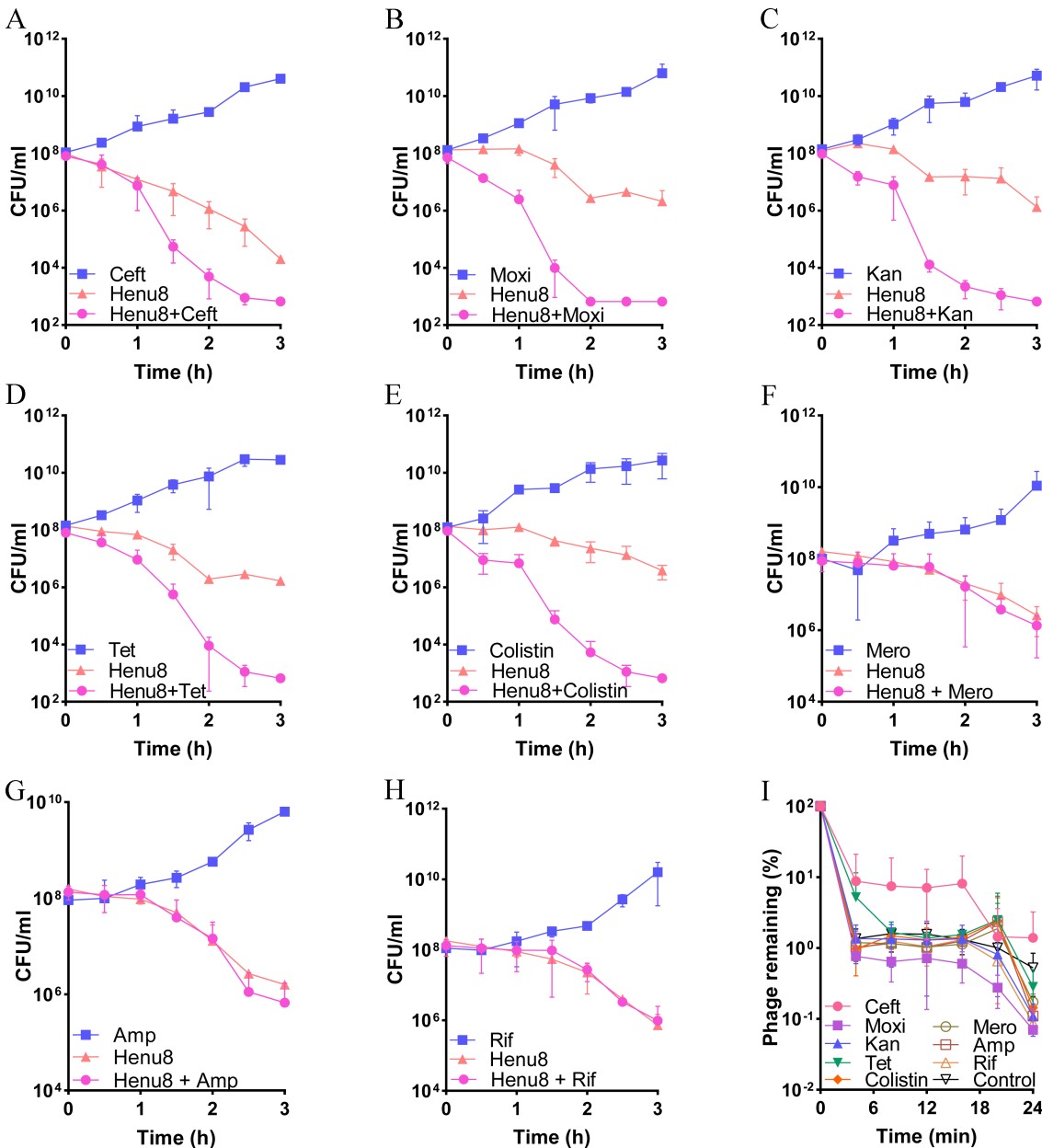

**FIG 7** Assessment of the synergistic potential of the bacteriophage Henu8 with antibiotics. *E. coli* BW25113 at $10^8$ CFU/mL was treated with a combination of antibiotics and the bacteriophage Henu8 collaboratively or individually for 3 h. Time-killing curves were plotted via the CFU counting method at the indicated times. The antibiotics used included 1/2 × MIC concentrations of ceftriaxone (Ceft, 0.0625 µg/mL) (A), moxifloxacin (Moxi, 0.0625 µg/mL) (B), kanamycin (Kan, 2 µg/mL) (C), tetracycline (Tet, 2 µg/mL) (D), polymyxin E (Colistin, 0.25 µg/mL) (E), meropenem (Mero, 0.0625 µg/mL) (F), ampicillin (Amp, 8 µg/mL) (G), and rifampicin (Rif, 8 µg/mL) (H), with bacteriophage Henu8 at an MOI of 0.01. All experiments were performed in triplicate individually. (I) Adsorption interference by antibiotics at 1/2 × MIC. The bacteriophage Henu8 was mixed with an *E. coli* BW25113 suspension at an MOI of 0.01. The adsorption rate was analyzed by comparison with the titer of the control group (without antibiotic treatment).

a novel species in the genus *Hanrivervirus* of the subfamily *Tempevirinae*, the bacteriophage Henu8, which can specifically lyse *E. coli* BW25113 and some clinical isolated MDR *E. coli*, broadening the script of *E. coli* bacteriophages as alternatives to combat chronic infection and bacteremia induced by *E. coli*.

Since biofilms act as protective shields for bacterial communities, antibiofilm ability is imperative for phages with therapeutic potential. The attachment of biofilm-encapsulated *E. coli* to the surface of medical apparatuses and instruments has long been

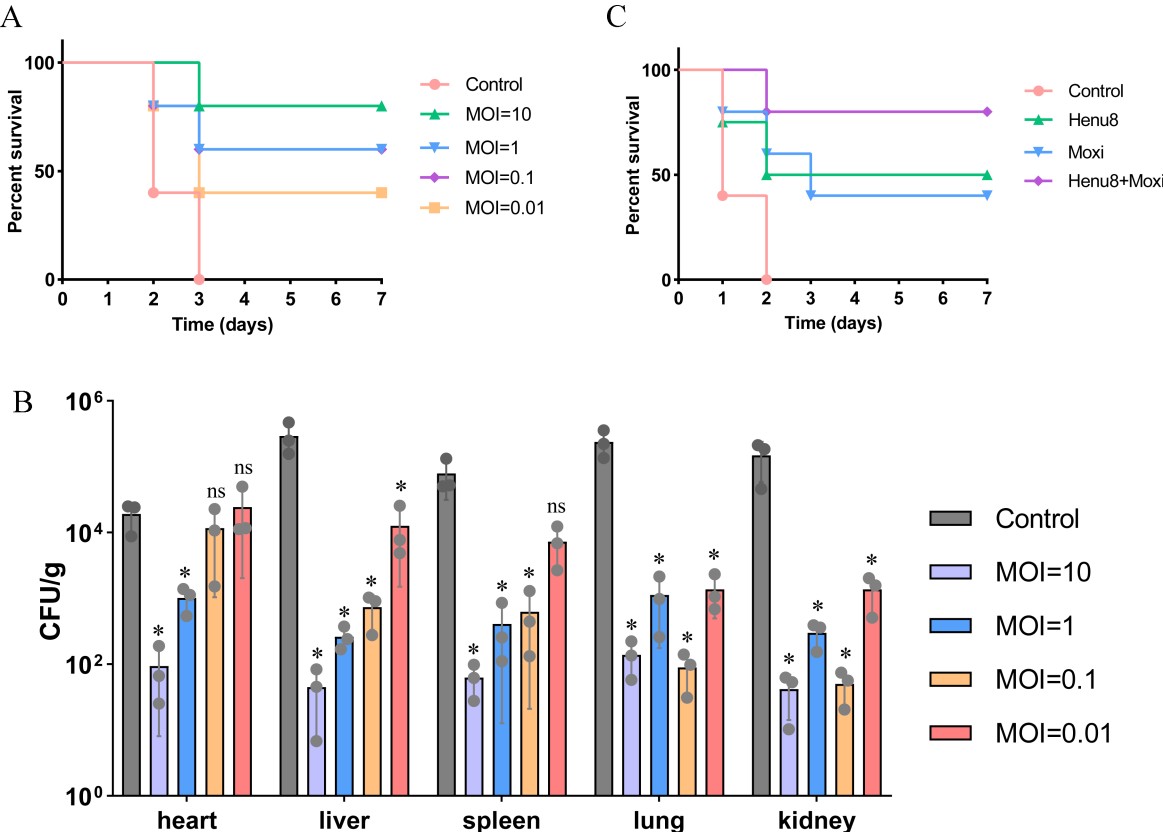

**FIG 8** Phage therapy of bacteriophage Henu8 in a C57BL/6N mouse *E. coli* BW25113 bacteremia model. (A) Survival rate analysis of phage therapy. Mice infected with *E. coli* BW25113 at $5 \times 10^9$ CFU/mL received phage therapy at different MOIs (10, 1, 0.1, and 0.01), which were observed and recorded for 7 days. (B) Bacterial loads of different organs in mouse bacteremia model. Mice were infected with $5 \times 10^8$ CFU/mL *E. coli* BW25113 via intraperitoneal injection. After 30 min, the infected mice were treated with Henu8 at different MOIs. The bacterial loads were determined via CFU counting in the dissected and ground tissues. (C) Survival rate analysis of the bacteriophage Henu8 in combination with moxifloxacin to treat bacteremic mice. Mice infected with *E. coli* BW25113 at a CFU/mL of $5 \times 10^9$ received the bacteriophage Henu8 (MOI = 0.01) and moxifloxacin (1.25 mg/kg), which were observed and recorded for 7 days.

recognized as the prime culprit behind infection by *E. coli* (38, 39). Thus, evaluations of biofilm disruption and biofilm inhibition ability are imperative to estimate its future clinical potential. However, the bacteriophage Henu8 exhibited effective biofilm eradication ability and inhibition ability, suggesting its usage for refractory infections induced by *E. coli* biofilms. The outstanding performance of bacteriophage Henu8 was also verified by the graphs of biofilms stained by CV, which, to some extent, further undeniably proved the high disruption efficiency of the bacteriophage Henu8 against *E. coli* biofilms. Some scholars have already noted the scenario of plaques on a double-layer agar plate, where there is a translucent halo circling the transparent plaques. Further experiments confirmed the existence of depolymerase, which plays a dominant role in degenerating the polysaccharide shell secreted by bacteria (40, 41). With further improvements in bioinformatics analysis, depolymerases have been identified as tail fiber proteins and tail spike proteins, as verified in dozens of reports (42–45). Since phages can be applied in infection treatment, the depolymerase of phages could be employed in clinical treatment because of its high eradication ability and low-tolerance frequency and purity without virulence interference, which has been supported by many *in vivo* scientific reports (46–48). In our study, the bacteriophage Henu8 also exhibited a narrow halo surrounding the transparent plaque, and bioinformatics analysis revealed the existence of ORFs 34, 42, and 51, which encode the tail fiber protein. Whether these genome sequences exhibit depolymerase activity, selectivity, or specificity, and whether purified protein solvents can be applied clinically remain unknown, and

further experiments are needed for verification. In addition, ORF 57, which encodes the HNH endonuclease according to bioinformatics analysis, may provide novel insight into antibacterial agent development since HNH endonucleases are considered novel adjuvants of antibiotics because of their ability to make bacteria more susceptible and easier to be eradicated (49).

The biological characteristics of phages may determine their potential in treating infection, where high burst sizes, short latent periods, and strong lytic abilities are favorable traits for phages used clinically. The bacteriophage Henu8 exhibited high burst sizes, extremely short latent periods, and strong lytic ability. These characterizations paved the way for further clinical usage. The bacteriophage Henu8 showed considerable inhibition ability within 4 h. However, with the occurrence of off-target effects and phage resistance, no visible inhibitory effect was observed after 4 h, as indicated by the increase in absorbance at $OD_{600}$, which suggested the brief acquisition of resistance, limiting its sole usage to combat bacterial infection. As reported in many articles, dozens of bacteriophages are faced with resistance from host bacteria, where bacteria go through the process of eradication first; then, resistance by multiple strategies obtained from the evolution with bacteria, such as halting the binding of phages, restriction-modification, the CRISPR-Cas9 system, and abortive infection, prevents the process of successful infection and cell lysis (50–52). To overcome the barriers associated with the clinical use of bacteriophages against MDR strains, a few "bacteriophage cocktails" have been designed. Dozens of lytic bacteriophages with diverse hosts and targets were included in the cocktail, exerting a broad spectrum of host specificity while causing minimal damage to normal nontarget cells except for inflammatory responses, the effectiveness of which has been verified by abundant reports of the biocontrol of *Enterococcus*, *Acinetobacter*, and other bacteria (18, 52–54). The potential negative effects of neither the low treatment effect induced by the interaction of different phages nor adverse effects such as inflammation caused by phage antigens contribute to long recovery periods, among which the strong lytic effect of phage cocktails may add to the release of endotoxin and aggravate the situation (52, 55).

Phages can inhibit bacterial growth, serving as an alternative to handle drug-resistant bacteria. However, the acquisition of phage resistance was easy, as previously reported (32). In our study, this phenomenon was verified by the increase in the $OD_{600}$ of the bacteriophage Henu8 after 4 h of coincubation. Therefore, to counter phage resistance to a single species of phage, phage cocktails may serve as a prominent choice by combining dozens of bacteriophages specific to bacteria of certain spectra (56). Although phage cocktails seem to perform well, inevitable and diverse phage resistance is accompanied by lipopolysaccharide (LPS) mutation, resulting in the inefficiency of the antibacterial strategy, as verified by medical application by Li et al. (57, 58). While the phage cocktails have been used in clinical trials to combat multidrug-resistant gram-negative bacteria and multidrug-resistant *Mycobacterium abscessus*, the phage-antibiotic combination also has the potential for eliminating the dissemination of MDR bacteria (59). The undeniable advantage of the phage-antibiotic combination lies in the low frequency of acquisition of both phage resistance and antibiotic resistance since several cases in which MDR bacteria regain susceptibility to antibiotics or a more significant eradication efficacy in clinical treatment have been reported (60–63). There has been a "genetic trade-off" theory explaining the acquisition of susceptibility to antibiotics, whereby by cotreating the bacteria with phages, the phage would adsorb to the targets, usually efflux pumps (30, 64). Efflux pumps are crucial proteins for bacteria to survive the pressure driven by antibiotics by pumping the antibiotics out of the bacteria, alleviating the pressure that high-concentration drugs bring inside bacteria. Occasionally, when treated with phages, bacteria are able to mutate in the expression of efflux pumps occupied by phages to alleviate pressure on the phages and avoid lysis (65, 66). In this way, the efflux pumps can reduce the concentration of drugs inside the cells, resulting in consequent cell death. In addition, a few research groups have verified the synergetic ability of several antibiotics against various bacterial strains, among which β-lactams and

quinolones have been verified the most (67–69). However, the synergetic effects of *E. coli* phages have not yet been sufficiently elucidated, according to several limited reports (66, 70, 71). Therefore, since the bacteriophage Henu8 has significant activity against growing bacteria and biofilms, whether it can be applied in a synergetic way with antibiotics is an interesting issue that remains to be determined. Here, we adopted several representative antibiotics from different classes to preliminarily verify their collaborative effect with Henu8. Several antibiotics indeed exhibited synergetic effects with phage Henu8, including ceftriaxone, moxifloxacin, kanamycin, tetracycline, and polymyxin E, preliminarily suggesting that these antibiotics may interfere with the biological behavior of the bacteriophage Henu8. In addition, when the adsorption rate was tested under 1/2 × MIC antibiotics, no significant differences were detected compared with that of the control group, except for the group treated with ceftriaxone sodium, which exhibited 10% interference with the adsorption of the bacteriophage Henu8, suggesting that the addition of antibiotics did not interfere with the adsorption of the bacteriophage Henu8 to bacteria in most cases. According to previous reports, several antibiotics can inhibit adsorption, which can likely be attributed to the common targets shared by antibiotics and bacteriophages (72–74). The synergetic potential against biofilms has also been reported in several other reports, where biofilm masses were tremendously reduced in *P. aeruginosa*, *S. aureus*, and *E. coli* (69, 71, 75). The probable mechanism lies in the primary destruction of biofilms and the formation of tunnels, where antibiotics can, therefore, easily penetrate the shield and play a role. Although the bacteriophage Henu8 exhibited prominent biofilm inhibition and destruction ability, whether it can terminate bacteria inside biofilms in a synergetic way remains unclear, and additional studies are needed to verify its efficacy. In summary, although there have been many insights into the synergistic effects of bacteriophages and antibiotics, the underlying synergetic mechanism requires further experiments, including additional antibiotics of the same class being tested, mutation analysis, and further molecular analysis. However, the bacteriophage Henu8 indeed broadened the script of the synergetic effect of *E. coli* phage with antibiotics, providing convincing materials for future research. In addition to the prevention of the antagonistic effects of the combination, the efficacy of the *in vivo* combination of phage and antibiotics also requires additional experiments for validation.

The ability to combat bacteria *in vivo* is considered the key to determining its clinical use. C57BL/6N mice are ideal candidates for mimicking bacteremia in mammals (65). Bacteria at high concentrations are widely implanted in the abdominal cavity and are absorbed by the peritoneum, thus forming a severe infection model. By closely observing the response and death rates of the mice, we could assess the therapeutic outcome. The bacteriophage Henu8 treatment reversed the survival rate by 40%–80% with increasing MOIs, and there could be more chances to witness a surge in survival since the *in vivo* performance was positively correlated with the amount of bacteriophage. When the bacteriophage Henu8 had an MOI = 0.01, the survival rate of the mice was only 40% or 50%, whereas when the phage was combined with moxifloxacin, the survival rate reached 80%. Treatment with the bacteriophage Henu8 also significantly reduced the bacterial loads in various organs of the mice. These results indicated that the bacteriophage Henu8 indeed has the ability to eradicate the bacteria *in vivo*. Together with the survival rate analysis, the *in vivo* experiments suggested that the bacteriophage Henu8 could be applied to mammalian *E. coli* infection. Since phages are immunogenic, there are cases in which patients are exposed to light or mild immunoreaction. Although dozens of clinical reports have verified the efficacy of antibiotic-phage therapy against MDR strains, opening a new chapter of antibacterial agents, a rational combination of specific phages with low immunogenicity should be considered and personalized (61, 62, 76). However, the tolerance of the bacteriophage Henu8 still needs further experimentation.

## Conclusion

In this study, the bacteriophage Henu8, which has strong lytic activity against *E. coli*, was isolated, purified, and characterized. In summary, the bacteriophage Henu8 has a large burst size, short latent period, and high adsorption rate to host bacterial strains, contributing to its antibacterial ability. High tolerance to extreme pH, UV destruction, and organic solvents increased the tolerance of bacteriophage Henu8 to environmental destruction. The bacteriophage Henu8 exhibited prominent antibacterial activity in both *in vitro* and *in vivo* bacteremia models, indicating its clinical potential in treating antibiotic-resistant *E. coli*. In addition, the bacteriophage Henu8 exhibited synergetic potential with several antibiotics. Thus, the identification of the bacteriophage Henu8 not only introduced a novel member of the *E. coli* phage but also expanded our horizons of phage-antibiotic combination usage and phage therapy against *E. coli* infection.

## MATERIALS AND METHODS

### Bacterial strains and culture conditions

All *E. coli* strains used in this study were from laboratories or were isolated from hospitals and stored at −80°C with 20% glycerol (vol/vol) in our laboratory. All bacterial strains were cultivated in 2× yeast extract-tryptone (YT) broth and incubated at 37°C and 200 rpm. PFU titration of phage was measured via a double-layer agar method, and bacterial CFUs were detected on 1.5% solid agar media.

### Isolation and purification of the bacteriophage Henu8

After centrifugation at 12,000 × g for 10 min, the supernatants containing the bacteriophage Henu8 were isolated from the sewage by filtering the samples through a 0.45 μm filter unit (NEST). By mixing the filtrate collected with the *E. coli* BW25113 suspension in 2 × YT broth, the phage was successfully enriched at 37°C at 200 rpm overnight. The supernatants were subsequently harvested via centrifugation at 12,000 × g for 10 min, after which double-layer agar plates were prepared to allow the formation of a single plaque unit. Once a single plaque was formed and isolated, dozens of rounds of enrichment and purification were initiated to obtain pure phage cultures.

### Enrichment of bacteriophage

High-titer bacteriophage was acquired by mixing a bacteriophage sample (100 μL) with fresh 2 × YT broth (100 mL) incubated with *E. coli* BW25113 in the mid-log phase. After overnight incubation at 37°C and 200 rpm, the mixture was centrifuged at 12,000 × g at 4°C for 10 min before the supernatants were filtered through 0.45 μm filters. The harvested supernatants were stored at 4°C for future use. Titration was detected via the double-layer agar method at each round of enrichment.

### Morphology of the bacteriophage Henu8

To observe the morphological traits of the bacteriophage Henu8, a double-layer agar plate was prepared according to a previously described method, where 0.7% agar containing phage was above the agar medium of 1.5%. After incubation at 37°C for 12 h, the morphology of the plaques was photographed. Individual phage spots were transferred to 1× PBS for transmission electron microscopy imaging. Transmission electron micrographs of the phage were taken with a JEM1200EX by ZHONG KE BAI CE (China, Beijing).

### Host range analysis

The ability of Henu8 to infect different bacteria was determined via plaque assays. To test the host range of the bacteriophage Henu8, clinical isolates and standard strains

(*E. coli*, *K. pneumoniae*, *P. aeruginosa*, and *A. baumannii*) were used. Briefly, bacteria incubated overnight were inoculated in fresh 2× YT broth medium at a ratio of 1:100 and further incubated at 37°C and 200 rpm until the final $OD_{600}$ was approximately 0.3. Then, double-layer agar was prepared from the strains mentioned above and 0.7% agar at a ratio of 1:10. The bacteriophage Henu8 (approximately $10^{10}$ PFU/mL) was serially diluted before being deposited onto the double-layer agar. The plates were incubated at 37°C for 12 h to fully allow the formation of plaques. Strains positive for infection were identified on the basis of the obvious formation of plaques. However, strains that were negative for infection were negative.

## Optimal multiplicity of infection

The MOI, estimated via gradient dilution and the soft agar overlay method, represents the ratio of bacteria to bacteriophage during infection. Briefly, bacteriophages of different titers ($10^9$, $10^8$, $10^7$, $10^6$, $10^5$, and $10^4$ PFU/mL) were added to the bacteria ($10^8$ CFU/mL) to obtain MOIs of 10, 1, 0.1, 0.01, 0.001, and 0.0001. After incubation at 37°C and 200 rpm for 2 h, samples of the mixtures were taken and centrifuged at 12,000 × g for 5 min, followed by serial dilution and plating on double-layer agar. After incubation at 37°C for 12 h, plaque counting was conducted once the incubation was finished.

## Adsorption of bacteriophages

Overnight-incubated bacteria were inoculated in 10 mL of 2× YT broth at a ratio of 1:100 and incubated at 37°C and 200 rpm until the $OD_{600}$ reached 0.3–0.4. Next, the bacteriophages were mixed with the bacteria to reach an MOI of 0.01 and incubated at room temperature for 15 min to facilitate the adsorption of the bacteriophage. Samples of 500 µL mixture were taken at intervals of 1 min, followed by centrifugation at 12,000 × g at 4°C for 2 min. After centrifugation, 100 µL supernatants of samples were aspirated into 900 µL saline to conduct serial dilutions and plated on double-layer agar. Phage plaques were counted after incubation at 37°C for 12 h.

## One-step growth curve

Overnight bacterial cultures were transferred into 10 mL of fresh 2× YT broth followed by incubation at 37°C at 200 rpm to reach the mid-log phase ($5 \times 10^9$ CFU/mL). After incubation, the bacteria were centrifuged at 8,000 rpm for 5 min and resuspended in 10 mL of 2× YT broth. Immediately after mixing with 10 mL of bacteriophage at an MOI of 0.01, the mixture was incubated at room temperature for 15 min to allow for adsorption. Next, the mixture was centrifuged at 8,000 rpm for 10 min to separate unadsorbed phages from the cells that were successfully infected. The pellets were resuspended in 1 mL of 2× YT broth and transferred to 20 mL of fresh medium. Samples were taken at intervals of 5 min (up to 30 min) and 10 min (up to 80 min). All the samples were centrifuged at 12,000 × g for 2 min before being serially diluted, plated, and incubated. Phage plaque counting was conducted after incubation for 12 h, and the PFU per milliliter was subsequently calculated.

## Chloroform stability, diethyl ether stability, and pH stability

To determine the stability of phage Henu8 in different solutions, the infectious capability of the bacteriophage was tested after treatment with chloroform and diethyl ether. One hundred microliter of bacteriophage ($10^{10}$ PFU/mL) was pipetted into tubes containing either 900 µL of 2× YT broth or the solutions mentioned above and incubated for 60 min at room temperature. After incubation, phage titration was performed on double-layer agar, and the mixture was further incubated at 37°C for 12 h until the formation of visible phage plaques. The stability of the bacteriophage at different pH values was also determined following the above method under saline solutions with different pH values (pH range, 1–12).

## UV sensitivity test of the bacteriophage Henu8

To demonstrate the resistance of the bacteriophage to UV light, bacteriophage Henu8 was irradiated with UV rays for 30 min (30W). In brief, samples were taken every 5 min, followed by serial dilution and further plating on double-layer agar. Visible phage plaques were counted after the incubation at 37°C for 12 h.

## Phage DNA isolation and sequencing

The genomic DNA of the bacteriophage Henu8 was extracted via a previously described method. The DNA quality was checked, and the DNA was subsequently sent to Shanghai Personal Biotechnology Co., Ltd. (China, Shanghai) for sequencing. The remaining DNA samples were digested with *Xba* I and *EcoR* I for agarose gel electrophoresis.

## Bioinformatics analysis

ORFs were predicted via Softberry (http://www.softberry.com/). The predicted protein functions of the ORFs were annotated via the BLASTp of the National Center for Biotechnology Information server with the nonredundant sequence database (77). The genome mapping and restriction sites were analyzed and drawn by SnapGene. Phylogenetic analyses based on the major capsid protein, putative portal protein, and terminase large subunit were performed with MEGA7 via the neighbor-joining method and detected via the bootstrap method with 1,000 test repetitions (78).

## The antibacterial ability of bacteriophage Henu8 against *E. coli in vitro*

Single colonies of *E. coli* BW25113 were picked and incubated overnight at 37°C and 200 rpm. After inoculation in 50 mL of fresh 2× YT broth, the suspension was incubated to the early logarithm. Phage solutions with different titers were prepared and mixed with bacterial suspensions at certain MOIs. The $OD_{600}$ was closely detected every 30 min via a spectrophotometer for 6 h and further graphed with GraphPad Prism. The experiments were performed in triplicate individually to ensure their accuracy.

## Inhibition and destruction of *E. coli* biofilms

For the testing inhibition of biofilms by bacteriophage Henu8, overnight-grown cultures of *E. coli* BW25113 were inoculated in fresh 2× YT broth and incubated at 37°C and 200 rpm for 2 h until the $OD_{600}$ value reached 0.4–0.5. The cultures were subsequently diluted 100 times in fresh 2× YT broth, which had a density of approximately $10^6$ CFU/mL. One hundred microliter of diluted medium was aspirated into a 96-well plate, followed by the pipetting of 100 μL of the bacteriophage Henu8 mixture, attaining the final MOIs of each well (10, 1, 0.1, 0.01, 0.001, and 0.0001). Next, the plates were statically incubated at 37°C for 24 h to generate biofilms. A total of 100 μL of diluted bacterial medium combined with 100 μL of saline was used as the control. The wells without any mixture were marked as the blank group. After 24 h, the wells were washed carefully with 1× PBS twice without damaging the biofilms on the bottom of the wells. After washing, the supernatants were discarded and refilled with 200 μL of methanol, after which the mixture was further stored at room temperature for 20 min. After fixation, the methanol was discarded, and the mixture was washed with 1 × PBS twice. Next, 0.1% CV (G1059, Solarbio) was used to dye the remaining biofilm for another 20 min at room temperature. After the completion of the dye, the CV was discarded, and the mixture was washed with 1× PBS twice until the supernatants were clear without dye. One hundred microliter of 33.3% (vol/vol) glacial acetic acid was used to dissolve the dyed biofilm, and the $OD_{570}$ was subsequently determined.

To test the destruction of biofilms by bacteriophage Henu8, bacteria with an $OD_{600}$ value of 0.3–0.4 were diluted 100-fold in fresh 2× YT medium. Then, 200 μL of the dilution mixture was placed in a new 96-well plate and statically incubated at 37°C for 20–24 h to allow the formation of biofilms. Next, the 2× YT broth in the plates was

discarded, followed by washing with 1× PBS on the wells containing biofilms. Then, 200 µL of bacteriophage was pipetted into the wells and coincubated at 37°C for another 20–24 h. After the completion of coincubation, the supernatants in the wells were discarded, and the bacteria were washed twice with 1× PBS without damaging the biofilms at the bottom. The steps used to quantify the biofilms were the same as those described above.

## Synergistic killing potential of antibiotics *in vitro*

In summary, single colonies of *E. coli* BW25113 were inoculated in 2× YT broth for incubation until the $OD_{600}$ reached 0.2 ($10^8$ CFU/mL). The phage suspension was diluted in saline to a PFU of $10^6$ per millimeter and pipetted into a bacteria suspension in combination with antibiotics at $1/2 \times$ MIC (ceftriaxone, moxifloxacin, kanamycin, tetracycline, polymyxin E, meropenem, ampicillin, and rifampicin). The remaining bacteria were calculated at 30 min intervals over a total period of 180 min. The bacteria were plated on a 1.5% 2× YT agar plate, and the number of bacterial colony-forming units was recorded overnight after incubation at 37°C.

## Adsorption of Henu8 by antibiotics

The adsorption rate was evaluated in the presence of various antibiotics at $1/2 \times$ MIC (ceftriaxone, moxifloxacin, kanamycin, tetracycline, polymyxin E, meropenem, ampicillin, and rifampicin). The method was modified and is described below (72). An *E. coli* BW25113 suspension of $10^8$ CFU/mL was mixed with bacteriophage Henu8 at $10^6$ PFU/mL to reach an MOI of 0.01. The antibiotics were subsequently aspirated into the suspension to attain the $1/2 \times$ MIC. Then, samples were taken at 4 min intervals and centrifuged at $12,000 \times g$ at 4°C for 5 min to collect the supernatants containing the phages. After serial dilution, adsorption was analyzed on a double-layer agar plate.

## *In vivo* phage treatment for *E. coli* infection

Briefly, bacteria incubated overnight were reinoculated in 2× YT broth and incubated for 2 h at 37°C and 200 rpm, attaining an $OD_{600} = 0.5$. After centrifugation at $8,500 \times g$ for 5 min, the pellets were resuspended and concentrated 10 times in saline. The bacteria were inoculated by intraperitoneal injection for infection purposes in 30 female C57BL/6N mice. Solutions of phages with different MOIs (10, 1, 0.1, and 0.01) were prepared and injected intraperitoneally, with each group containing five mice, where saline injection served as a negative control. The survival rate of the mice infected with bacteria was monitored and recorded for the next 7 days. To clarify the therapeutic efficacy of bacteriophage Henu8 in septicemic mice in terms of the bacterial load in various organs, we conducted an experiment similar to that of survival rate analysis. In brief, when the $OD_{600}$ value of bacterial suspension in 2× YT broth reached 0.3, the bacterial suspension was centrifuged at $8,500 \times g$ for 5 min, resuspended in saline, and injected intraperitoneally into the mice. Phage therapy was performed 30 min after infection through the injection of phage solution diluted in saline to attain certain MOIs (10, 1, 0.1, and 0.01) *in vivo*. Forty-eight hours after therapy, colony-forming units in certain organs (heart, liver, spleen, lung, and kidney) were evaluated on 1.5% 2× YT agar plates. The data are presented in the measurement as CFUs per gram.

## ACKNOWLEDGMENTS

The present study is supported by the Natural Science Foundation of Henan (202300410052); the China Postdoctoral Science Foundation (2020M682279); Projects for College Students in Henan University (XJ2024333, S202410475020); Kaifeng Science and Technology Development Program Projects (2303007); Henan Science and Technology Development Program Projects (242102310023); and Provinces and ministries jointly built key projects of Henan Provincial Health Commission (SBGJ202402084 and SBGJ202302089); The platform construction fund of Henan Province EngineeringTechnology Research Center of Rapid-Accuracy Medical Diagnostics (30389).

F.Z., K.W., S.J., X.L., and Q.L. carried out the experiments and analyzed the data. T.T. analyzed the genomic information of the phage. K.W., W.Z., and Q.L. drafted the manuscript. L.W. and Q.L. conceived and designed the study, revised the manuscript, and provided research funding. All the authors have read and approved the final manuscript.

## AUTHOR AFFILIATIONS

[1]Henan Province Engineering Technology Research Center of Rapid-Accuracy Medical Diagnostics, Department of Clinical Laboratory, The First Affiliated Hospital of Henan University, Henan University, Kaifeng, China

[2]The Joint National Laboratory of Antibody Drug Engineering, Henan University, Kaifeng, China

[3]Department of Microbiology, College of Basic Medical Sciences, Henan University, Kaifeng, China

## AUTHOR ORCIDs

Li Wang ⓘ http://orcid.org/0009-0001-1822-5887
Qiming Li ⓘ http://orcid.org/0000-0002-6916-3954

## FUNDING

| Funder | Grant(s) | Author(s) |
| --- | --- | --- |
| Natural Science Foundation of Henan Province (Henan Natural Science Foundation) | 202300410052 | Qiming Li |
| China Postdoctoral Science Foundation | 2020M682279 | Qiming Li |
| Projects for cellege students in Henan university | XJ2024333, S202410475020 | Wenwen Zhang |
| Kaifeng science and technology development program | 2303007 | Qiming Li |
| Henan science and technology development program projects | 242102310023 | Qiming Li |
| Provinces and ministries jointly built key projects of henan provincial health commussion | SBGJ202402084, SBGJ202302089 | Li Wang |
| The platform construction fund of Henan Province EngineeringTechnology Research Center of Rapid-Accuracy Medical Diagnostics | 30389 | Li Wang |

## AUTHOR CONTRIBUTIONS

Fang Zhou, Data curation, Investigation, Software, Writing – original draft, Writing – review and editing | Kexiao Wang, Data curation, Formal analysis, Writing – original draft, Writing – review and editing | Shuai Ji, Data curation, Writing – review and editing | Xiaochen Liao, Data curation, Writing – review and editing | Wenwen Zhang, Data curation, Writing – review and editing | Tieshan Teng, Writing – review and editing | Li Wang, Funding acquisition, Project administration, Writing – review and editing | Qiming Li, Conceptualization, Data curation, Funding acquisition, Project administration, Writing – original draft, Writing – review and editing

## ETHICS APPROVAL

The study was approved by the Ethics Committee of Henan University. All procedures were conducted following the relevant guidelines.

## ADDITIONAL FILES

The following material is available online.

## Supplemental Material

**Figure S1 to S4 and Table S1 (Spectrum01633-24-s0001.doc).** Clinical isolate data.

## Open Peer Review

**PEER REVIEW HISTORY (review-history.pdf).** An accounting of the reviewer comments and feedback.

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
