## [Reviewer comments · Microbiology Spectrum]

Microbiology Spectrum

The virulent bacteriophage Henu8 as an antimicrobial synergist against *Escherichia coli*

qiming li, fang zhou, kexiao wang, shuai ji, Xiaochen Liao, Tieshan Teng, Li Wang, and wenwen zhang

Corresponding Author(s): qiming li, Henan University

Review Timeline:

Submission Date:	July 3, 2024
Editorial Decision:	December 10, 2024
Revision Received:	February 24, 2025
Editorial Decision:	March 14, 2025
Revision Received:	March 21, 2025
Accepted:	April 2, 2025

Editor: David Pride

Reviewer(s): Disclosure of reviewer identity is with reference to reviewer comments included in decision letter(s). The following individuals involved in review of your submission have agreed to reveal their identity: Juhee Ahn (Reviewer #1); Victor Gonzalez (Reviewer #3)

Transaction Report:

DOI: <https://doi.org/10.1128/spectrum.01633-24>

Re: Spectrum01633-24 (**Virulent bacteriophage Henu8 as an antimicrobial synergist against *Escherichia coli***)

Dear Dr. qiming li:

Thank you for the privilege of reviewing your work. Below you will find my comments, instructions from the Spectrum editorial office, and the reviewer comments.

Revision Guidelines

Sincerely,
David Pride
Editor
Microbiology Spectrum

Reviewer #1 (Comments for the Author):

This study clearly outlines the significance of *E. coli* in health care and the challenge of antibiotic resistance, effectively justifying the exploration of bacteriophages as a therapeutic alternative. The newly isolated phages are well characterized in this study.

There are few minor comments as follows;

- L236: balancing

- L 251: act as protective shields for
- L257: nice to effective
- L281: splendid to considerable
- L292: Revise the clumsy phrase - by scholars confronting ...
- L301: Phage cocktails can cause much more diverse emergence of phage-resistant bacteria than single phage treatment.
- L312: germs to bacteria
- Discussion - The optimum MOI result should be further discussed, why MOI=0.01 showed most lytic activity, rather than 0.1, 1 or higher MOIs and why the optimum MOI was different in vivo study.
- All experiments were conducted with BW25113. But, the antimicrobial activity of phages against MDR strains should be more interested, then antimicrobial activity should have tested with MDR strains.

Reviewer #3 (Comments for the Author):

The manuscript by Zhou et al. reports on the characterization of the phage Henu8 of E coli. The authors proposed this phage is highly virulent and acts synergistically with antibiotics to eliminate E. coli in vitro and in vivo. The manuscript has two parts: microbiological characterization and genome sequence of the phage, as well as virulence testing and evaluation of its performance in biofilm. In some of these aspects, the manuscript seems to lose the focus. Although the first part is necessary to perform, it is less valuable than the biological assays of the phage. However, several points in the experiment should be clarified to contribute to the field. I hope the following comments help to improve the manuscript.

Abstract.

I suggest reviewing the abstract to highlight the relevant scientific findings, not enumerate the results.

Details:

l.8 "Henu8, which could selectively lysate E. coli". Henu8 can lysate the group of E. coli strains tested.

l. 20 The "translucent halo is not visible in Fig. 1. Can it provide a better image and a close-up of the plaque?"

l. 21 It seems that the burst size is too large. Provide details on calculating this and similar examples in the literature with such a high burst size.

l. 21 There are words such as "illuminate," "equipped," "exceptional," and "brilliant" throughout the text that is not of regular use in scientific publications. Review.

Introduction.

It is good, although excessive.

Results.

l. 115 "multiple purification," precise the number.

l.119 "average" of how many virions?

l. 120-121 A concise statement of the intended experiment could be understood in the writing.

l.125 "collected in our laboratory." The origin of the strains (environment, hospital) must be explicit. If this is the case, how were these strains isolated and characterized? Perhaps, I can suppose, strains are part of the laboratory collection.

Fig. 2. Why does the phage have a high titer at MOI 0.01 of about 2×10^8 (A), but the phage titer calculated in the one-step growth curve is 10^{12} ?

l. 138 What is the objective of testing the phage under different conditions? Unless some of the factors tested were likely found in physiological conditions or for maintaining the phage for a long time, this section seems unjustified.

l. Is Softberry software for gene annotation? Clarify.

l. 154-158. In my opinion, it is unnecessary to report on the restriction sites unless the objective of the manuscript is genetic modification. Moreover, the restriction enzyme was not correctly addressed: lack of the undigested control, and fragments are not in the correct size.

Fig. 6. It is unclear if the phage effects on "preformed biofilm" and "biofilm inhibition" are two different phenomena or the same. There are no distinctions between B and C plots (the letters are missing). It could be necessary to compare the columns side by side and obtain a statistic for the comparison. The term "aspirated" is not usual.

Fig. 7 What is the correlation between phage titers and biofilm formation and between phage titers and incubation time?

Fig. 8. How valid is the experiment if the Henu8 phage does not eliminate the cells? The authors show that after 4 hours of incubation, the remnant cells recover to growth. The experiment here is cut after three hours. Otherwise, explain why this incubation time was selected. I also see that this and the following experiments in mice are the most critical sections in the manuscript.

Discussion.

It should be limited once the focus of the manuscript is defined.

- Upload point-by-point responses to the issues raised by the reviewers in a file named "Response to Reviewers," NOT in your cover letter.
- Upload a compare copy of the manuscript (without figures) as a "Marked-Up Manuscript" file.
- Upload a clean .DOC/.DOCX version of the revised manuscript and remove the previous version.
- Each figure must be uploaded as a separate, editable, high-resolution file (TIFF or EPS preferred), and any multipanel figures must be assembled into one file.
- Any supplemental material intended for posting by ASM should be uploaded with their legends separate from the main manuscript. You can combine all supplemental material into one file (preferred) or split it into a maximum of 10 files with all associated legends included.

Thanks to the editorial team for the reminder, we made the changes exactly as requested.

Reviewer #1 (Comments for the Author):

This study clearly outlines the significance of E. coli in health care and the challenge of antibiotic resistance, effectively justifying the exploration of bacteriophages as a therapeutic alternative. The newly isolated phages are well characterized in this study. There are few minor comments as follows;

- L236: balancing
- L 251: act as protective shields for
- L257: nice to effective
- L281: splendid to considerable
- L292: Revise the clumsy phrase - by scholars confronting ...
- L301: Phage cocktails can cause much more diverse emergence of phage-resistant bacteria than single phage treatment.
- L312: germs to bacteria

Thank you sincerely for your constructive and precise suggestions! To enhance the readability and clarity, we've corrected the expression mistakes mentioned above.

- Discussion - The optimum MOI result should be further discussed, why MOI=0.01 showed most lytic activity, rather than 0.1, 1 or higher MOIs and why the optimum MOI was different in vivo study.

Thank you for your question! The hypothesis was that when MOI was set to be 10, 1, or 0.1, the lytic phenomenon would be too strong to give rise to the reproduction of phage, resulting in the less burst size and thus the less lytic manner on the whole. On the contrary, when the MOI was below the optimal MOI, the lytic process would be prolonged thus holding back the reproduction and release of regenerated virions. Therefore, the optimal MOI stands as the best ratio of virions and cells, allowing the highest yields and most prominent bioreaction.

Considering the difference between *in vitro* MOI and *in vivo* MOI, we put forward the hypothesis that intraperitoneal injection allows the swift shift of phage from the abdomen to the whole circular system due to the powerful absorption capacity of the peritoneum. Under this circumstance, the phage solution we injected, to some extent, was diluted by the blood circulation, leading to the reduction of phage titer in blood and organs. Besides, the distribution of bacteria was diversified, adding to the differences in the bacteria distribution and the final concentration of bacterial loads in reality. Therefore, the real MOIs in different organs may be

below the optimal MOI in *in vitro* experiments. In this way, the improvement of survival expectation is exerted in an MOI-dependent manner in *in vivo* experiments¹.

- All experiments were conducted with BW25113. But, the antimicrobial activity of phages against MDR strains should be more interested, then antimicrobial activity should have tested with MDR strains.

Thank you for your suggestions on testing the MDR strains of *E. coli*! As requested, we've isolated a multidrug-resistant strain of *E. coli* able to be lysed by Henu8 from the clinical laboratory of the First Affiliated Hospital of Henan University, named A57. To evaluate the antimicrobial activity of Henu8 against the strain, simultaneous detection of the changes of absorbance at 600nm was conducted by treating the A57 with Henu8 at indicated MOIs. The antibacterial capability was demonstrated as the halt in bacterial growth (Fig. S3).

Apart from the *in vitro* test of anti-MDR ability, the healing effect of Henu8 on the bacteremia mice model was also evaluated where an evident reversal of survival expectancy was observed in a concentration-dependent manner. As indicated, all mice died within 1 day of model establishment, while groups treated with phage Henu8 exerted profound antibacterial ability by elevating the survival rate to 40%, 40%, 40%, and 60% for MOIs at 0.01, 0.1, 1, and 10, respectively, providing solid evidence that Henu8 is capable of eradicating both BW25113 strain and MDR A57 strain (Fig. S4).

Reviewer #3 (Comments for the Author):

The manuscript by Zhou et al. reports on the characterization of the phage Henu8 of *E. coli*. The authors proposed this phage is highly virulent and acts synergistically with antibiotics to eliminate *E. coli* *in vitro* and *in vivo*. The manuscript has two parts: microbiological characterization and genome sequence of the phage, as well as virulence testing and evaluation of its performance in

biofilm. In some of these aspects, the manuscript seems to lose the focus. Although the first part is necessary to perform, it is less valuable than the biological assays of the phage. However, several points in the experiment should be clarified to contribute to the field. I hope the following comments help to improve the manuscript.

Abstract.

I suggest reviewing the abstract to highlight the relevant scientific findings, not enumerate the results.

Thank you for your suggestion! We've revised the abstract to make it more readable and concise instead of enumerating the results.

Details:

I.8 "Henu8, which could selectively lysate E. coli". Henu8 can lysate the group of E. coli strains tested.

Thank you for your dedicated correction. We've rephrased the sentence marked in red.

I. 20 The "translucent halo is not visible in Fig. 1. Can it provide a better image and a close-up of the plaque?"

Thank you for your pointing out the problem in the figure presented. We've placed the photographs shot at different times to exhibit the expansions both in plaques and translucent halos as well as to provide readers with a more apparent morphology.

I. 21 It seems that the burst size is too large. Provide details on calculating this and similar examples in the literature with such a high burst size.

Thank you for your dedicated review. That was our mistake to wrongly calculate the burst size of phage Henu8 when inserting the data into Excel. The mistake has been corrected from "2750" to "275" and marked in red.

I. 21 There are words such as "illuminate," "equipped," "exceptional," and "brilliant" throughout the text that is not of regular use in scientific publications. Review.

Thank you for pointing out the wording problems in our article! We've replaced the mentioned words with "demonstrate", "profound", "prominent", and "excellent".

Introduction.

It is good, although excessive.

Results.

I. 115 "multiple purification," precise the number.

Thank you for your dedicated review! After confirmation, the “multiple purifications” have been corrected as “5 rounds’ purifications”, by which we finally collected and stored the ideal concentration of phage Henu8 prepared for future experiments.

I.119 "average" of how many virions?

Thank you for pointing out that detail. The average diameter was estimated based on 3 virions of phage Henu8.

I. 120-121 A concise statement of the intended experiment could be understood in the writing.

Your suggestions are of great help to us! We’ve rephrased the statement in a more concise way to emphasize the crucial part of the experiments. Sentences marked in red were the revised version.

I.125 "collected in our laboratory." The origin of the strains (environment, hospital) must be explicit. If this is the case, how were these strains isolated and characterized? Perhaps, I can suppose, strains are part of the laboratory collection.

Thank you for your careful insights! We’ve ignored the clarity of strains tested in the lab by detailly present them in the table. As requested, we’ve reframed the table by adding the “isolation” and “source” into the original version to specify the strains’ information. Besides, the A57 E. coli strain isolated from the clinical laboratory of the First Affiliated Hospital of Henan University has been proven susceptible to phage Henu8 and tested in the following experiments and added to Table 1.

Fig. 2. Why does the phage have a high titer at MOI 0.01 of about 2×10^8 (A), but the phage titer calculated in the one-step growth curve is 1012?

Thank you for your dedicated review. Through meticulous review and validation of the original datasets, we detected and subsequently addressed certain inaccuracies in Figure 2c. The corrected version was marked in red and Fig 2C was graphed. Though the value still seems to be high, the probable reason lying in the phenomenon may be the change of experimental conditions and original CFU the mixture contains. Additionally, the observed discrepancies may be attributed to differences in experimental conditions, particularly in the one-step growth curve protocol where bacterial cultures were resuspended in fresh medium. Furthermore, variations in the incubation time allocated for bacterial adsorption could also contribute to these differences. Similar observations have been reported in other studies¹.

I. 138 What is the objective of testing the phage under different conditions? Unless some of the factors tested were likely found in physiological conditions or for maintaining the phage for a long time, this section seems unjustified.

Thank you for your detailed review. In response to your valuable suggestion, we have incorporated the phage stability analysis at various temperatures into the main text, while relocating the stability experiments involving UV light and organic reagents to the supplementary material for enhanced readability and focus.

I. Is Softberry software for gene annotation? Clarify.

Softberry serves as an online ORF prediction software. Instead, the functions of genes were predicted by BLASTp.

I. 154-158. In my opinion, it is unnecessary to report on the restriction sites unless the objective of the manuscript is genetic modification. Moreover, the restriction enzyme was not correctly addressed: lack of the undigested control, and fragments are not in the correct size.

Thank you for your suggestions! To enhance the functional annotation and biological relevance of our genomic analysis, we employed circular genome maps with comprehensive gene function

annotations, replacing the previous version that primarily highlighted enzymatic sites. Furthermore, through comprehensive restriction enzyme digestion analysis, we experimentally validated the circular conformation of the henu8 genome. We've also noticed the miss of "control". Thus, another electrophoresis with "control" was complemented to our article, and the fragments were carefully checked for credibility.

Fig. 6. It is unclear if the phage effects on "preformed biofilm" and "biofilm inhibition" are two different phenomena or the same. There are no distinctions between B and C plots (the letters are missing). It could be necessary to compare the columns side by side and obtain a statistic for the comparison. The term "aspirated" is not usual.

Thank you for your suggestions! We've added the missing mark to distinguish the "performed biofilm" and "biofilm inhibition". "Preformed biofilm" stands for the disruption ability of the biofilms already formed, where phage Henu8 was added to the wells 24h after the formation of bacterial biofilm, a shield is ready. On the contrary, "Biofilm inhibition" stands for the inhibition ability of the biofilm by co-treating the bacteria and phage Henu8 once the incubation began. The missing letters were added. As indicated, the inhibition test of Henu8 exerted a relatively strong anti-biofilm ability compared with that of the destruction test, though the differences are minor among different MOIs.

Fig. 7 What is the correlation between phage titers and biofilm formation and between phage titers and incubation time?

The lack of a linear correlation between phage titer and biofilm inhibition can be attributed to the repeated cycles of phage infection that occur during the incubation period with bacterial cells. In the growth curve analysis, no statistically significant differences were observed among the various phage titers. Furthermore, the one-step growth curve experiments revealed that bacterial populations rapidly developed phage resistance within a remarkably short timeframe, consequently eliminating any direct correlation between incubation duration and biofilm inhibition efficacy.

During our thorough analysis of the experimental data, we identified potential inconsistencies in the results presented in Figure 7. After careful consideration and rigorous verification, we have determined that these findings may not meet the stringent standards required for scientific publication. Consequently, we have decided to exclude this figure from the manuscript to ensure

the overall scientific integrity and clarity of our work. This editorial decision will enable readers to better comprehend the core findings and maintain the highest standards of scientific accuracy in our publication.

Fig. 8. How valid is the experiment if the Henu8 phage does not eliminate the cells? The authors show that after 4 hours of incubation, the remnant cells recover to growth. The experiment here is cut after three hours. Otherwise, explain why this incubation time was selected. I also see that this and the following experiments in mice are the most critical sections in the manuscript.

Thank you for the question. Most of the phages reported so far are susceptible to resistance *in vitro* but have good therapeutic effects *in vivo*². The reason may be that *in vitro* the bacteria are larger in number and prone to resistant mutations. *In vivo*, immune cells are helpful in the treatment of phages. Bacteria that develop resistance to phages do so at the cost of adaptive mutations and may therefore be extremely sensitive to antibiotics³. At the same time, antibiotics also inhibit bacteria from developing resistance to phages⁴. Therefore, phages and antibiotics have a synergistic effect in the bactericidal process. In our experimental observations, no bacterial growth (mutation production) was detected within the initial 8-hour period. A modest and irregular increase in bacterial count was observed after this time frame. Consequently, this study focuses exclusively on investigating the synergistic effects of henu8 in combination with antibiotics. We appreciate your insightful question, which has prompted us to consider further exploration of bacterial responses under the dual selective pressure of henu8 and antibiotics in our future research.

Discussion.

It should be limited once the focus of the manuscript is defined.

Thank you for suggestion. The full text has been completely overhauled in response to reviewer and editorial comments.

- 1 Han, P. *et al.* Characterization of the Bacteriophage BUCT603 and Therapeutic Potential Evaluation Against Drug-Resistant *Stenotrophomonas maltophilia* in a Mouse Model. *Front Microbiol* **13**, 906961, doi:10.3389/fmicb.2022.906961 (2022).
- 2 Han, P., Pu, M., Li, Y., Fan, H. & Tong, Y. Characterization of bacteriophage BUCT631 lytic for K1 *Klebsiella pneumoniae* and its therapeutic efficacy in *Galleria mellonella* larvae. *Virology* **38**, 801-812, doi:10.1016/j.virus.2023.07.002 (2023).
- 3 Cao, X. *et al.* Enhanced bacteriostatic effects of phage vB_C4 and cell wall-targeting antibiotic combinations against drug-resistant *Aeromonas veronii*. *Microbiology spectrum* **13**, e0190824, doi:10.1128/spectrum.01908-24 (2025).
- 4 Parab, L. *et al.* Chloramphenicol and gentamicin reduce the evolution of resistance to phage PhiX174 by suppressing a subset of *E. coli* LPS mutants. *PLoS biology* **23**, e3002952, doi:10.1371/journal.pbio.3002952 (2025).

Re: Spectrum01633-24R1 (**The virulent bacteriophage Henu8 as an antimicrobial synergist against *Escherichia coli***)

Dear Dr. qiming li:

Thank you for the privilege of reviewing your work. Below you will find my comments, instructions from the Spectrum editorial office, and the reviewer comments.

I am struggling so much with the following response to the review that I believe the authors should clarify this response before I send it out to the reviewers. Please clarify this response:

- Discussion - The optimum MOI result should be further discussed, why MOI=0.01 showed most lytic activity, rather than 0.1, 1 or higher MOIs and why the optimum MOI was different in vivo study.

Thank you for your question! The hypothesis was that when MOI was set to be 10, 1, or 0.1, the lytic phenomenon would be too strong to give rise to the reproduction of phage, resulting in the less burst size and thus the less lytic manner on the whole. On the contrary, when the MOI was below the optimal MOI, the lytic process would be prolonged thus holding back the reproduction and release of regenerated virions. Therefore, the optimal MOI stands as the best ratio of virions and cells, allowing the highest yields and most prominent bioreaction.

Considering the difference between in vitro MOI and in vivo MOI, we put forward the hypothesis that intraperitoneal injection allows the swift shift of phage from the abdomen to the whole circular system due to the powerful absorption capacity of the peritoneum. Under this circumstance, the phage solution we injected, to some extent, was diluted by the blood circulation, leading to the reduction of phage titer in blood and organs. Besides, the distribution of bacteria was diversified, adding to the differences in the bacteria distribution and the final concentration of bacterial loads in reality. Therefore, the real MOIs in different organs may be below the optimal MOI in in vitro experiments. In this way, the improvement of survival expectation is exerted in an MOI-dependent manner in in vivo experiments¹

Revision Guidelines

Sincerely,
David Pride
Editor
Microbiology Spectrum

- Upload point-by-point responses to the issues raised by the reviewers in a file named "Response to Reviewers," NOT in your cover letter.
- Upload a compare copy of the manuscript (without figures) as a "Marked-Up Manuscript" file.
- Upload a clean .DOC/.DOCX version of the revised manuscript and remove the previous version.
- Each figure must be uploaded as a separate, editable, high-resolution file (TIFF or EPS preferred), and any multipanel figures must be assembled into one file.
- Any supplemental material intended for posting by ASM should be uploaded with their legends separate from the main manuscript. You can combine all supplemental material into one file (preferred) or split it into a maximum of 10 files with all associated legends included.

Thanks to the editorial team for the reminder, we made the changes exactly as requested.

Reviewer #1 (Comments for the Author):

This study clearly outlines the significance of E. coli in health care and the challenge of antibiotic resistance, effectively justifying the exploration of bacteriophages as a therapeutic alternative. The newly isolated phages are well characterized in this study. There are few minor comments as follows;

- L236: balancing
- L 251: act as protective shields for
- L257: nice to effective
- L281: splendid to considerable
- L292: Revise the clumsy phrase - by scholars confronting ...
- L301: Phage cocktails can cause much more diverse emergence of phage-resistant bacteria than single phage treatment.
- L312: germs to bacteria

Thank you sincerely for your constructive and precise suggestions! To enhance the readability and clarity, we've corrected the expression mistakes mentioned above.

- Discussion - The optimum MOI result should be further discussed, why MOI=0.01 showed most lytic activity, rather than 0.1, 1 or higher MOIs and why the optimum MOI was different in vivo study.

Thank you for your question! When the MOI is too low, the number of phages is insufficient to infect all host bacteria, resulting in some bacteria remaining uninfected and reducing the overall lysis efficiency. When the MOI is too high, multiple phages may simultaneously infect the same bacterial cell, leading to "overinfection." In this scenario, phages compete for the host's resources, which can reduce the replication efficiency of individual phages. At the optimal MOI, the ratio of phages to host bacteria reaches a balance, enabling efficient infection of most bacterial cells while avoiding resource wastage^{1,2}.

In vitro, phages can directly interact with bacteria, making it easier for them to adsorb onto the bacterial surface. However, in *in vivo* experiments, infected bacteria can invade various tissues and organs. When using phages for treatment, the phages need to reach these tissues and organs to adsorb to and infect the bacteria. During this process, the number of phages that actually come into contact with the bacteria is reduced. Therefore, a higher phage concentration is required to achieve the optimal multiplicity of infection (MOI) of 0.01³.

- All experiments were conducted with BW25113. But, the antimicrobial activity of phages against MDR strains should be more interested, then antimicrobial activity should have tested with MDR strains.

Thank you for your suggestions on testing the MDR strains of *E. coli*! As requested, we've isolated a multidrug-resistant strain of *E. coli* able to be lysed by Henu8 from the clinical laboratory of the First Affiliated Hospital of Henan University, named A57. To evaluate the antimicrobial activity of Henu8 against the strain, simultaneous detection of the changes of absorbance at 600nm was conducted by treating the A57 with Henu8 at indicated MOIs. The antibacterial capability was demonstrated as the halt in bacterial growth (Fig. S3).

Apart from the *in vitro* test of anti-MDR ability, the healing effect of Henu8 on the bacteremia mice model was also evaluated where an evident reversal of survival expectancy was observed in a concentration-dependent manner. As indicated, all mice died within 1 day of model establishment, while groups treated with phage Henu8 exerted profound antibacterial ability by elevating the survival rate to 40%, 40%, 40%, and 60% for MOIs at 0.01, 0.1, 1, and 10, respectively, providing solid evidence that Henu8 is capable of eradicating both BW25113 strain and MDR A57 strain (Fig. S4).

Reviewer #3 (Comments for the Author):

The manuscript by Zhou et al. reports on the characterization of the phage Henu8 of *E. coli*. The authors proposed this phage is highly virulent and acts synergistically with antibiotics to eliminate *E. coli* *in vitro* and *in vivo*. The manuscript has two parts: microbiological characterization and genome sequence of the phage, as well as virulence testing and evaluation of its performance in biofilm. In some of these aspects, the manuscript seems to lose the focus. Although the first part is necessary to perform, it is less valuable than the biological assays of the phage. However,

several points in the experiment should be clarified to contribute to the field. I hope the following comments help to improve the manuscript.

Abstract.

I suggest reviewing the abstract to highlight the relevant scientific findings, not enumerate the results.

Thank you for your suggestion! We've revised the abstract to make it more readable and concise instead of enumerating the results.

Details:

I.8 "Henu8, which could selectively lysate E. coli". Henu8 can lysate the group of E. coli strains tested.

Thank you for your dedicated correction. We've rephrased the sentence marked in red.

I. 20 The "translucent halo is not visible in Fig. 1. Can it provide a better image and a close-up of the plaque?"

Thank you for your pointing out the problem in the figure presented. We've placed the photographs shot at different times to exhibit the expansions both in plaques and translucent halos as well as to provide readers with a more apparent morphology.

I. 21 It seems that the burst size is too large. Provide details on calculating this and similar examples in the literature with such a high burst size.

Thank you for your dedicated review. That was our mistake to wrongly calculate the burst size of phage Henu8 when inserting the data into Excel. The mistake has been corrected from "2750" to "275" and marked in red.

I. 21 There are words such as "illuminate," "equipped," "exceptional," and "brilliant" throughout the text that is not of regular use in scientific publications. Review.

Thank you for pointing out the wording problems in our article! We've replaced the mentioned words with "demonstrate", "profound", "prominent", and "excellent".

Introduction.

It is good, although excessive.

Results.

I. 115 "multiple purification," precise the number.

Thank you for your dedicated review! After confirmation, the "multiple purifications" have been corrected as "5 rounds' purifications", by which we finally collected and stored the ideal

concentration of phage Henu8 prepared for future experiments.

I.119 "average" of how many virions?

Thank you for pointing out that detail. The average diameter was estimated based on 3 virions of phage Henu8.

I. 120-121 A concise statement of the intended experiment could be understood in the writing.

Your suggestions are of great help to us! We've rephrased the statement in a more concise way to emphasize the crucial part of the experiments. Sentences marked in red were the revised version.

I.125 "collected in our laboratory." The origin of the strains (environment, hospital) must be explicit. If this is the case, how were these strains isolated and characterized? Perhaps, I can suppose, strains are part of the laboratory collection.

Thank you for your careful insights! We've ignored the clarity of strains tested in the lab by detailly present them in the table. As requested, we've reframed the table by adding the "isolation" and "source" into the original version to specify the strains' information. Besides, the A57 E. coli strain isolated from the clinical laboratory of the First Affiliated Hospital of Henan University has been proven susceptible to phage Henu8 and tested in the following experiments and added to Table 1.

Fig. 2. Why does the phage have a high titer at MOI 0.01 of about 2×10^8 (A), but the phage titer calculated in the one-step growth curve is 1012?

Thank you for your dedicated review. Through meticulous review and validation of the original datasets, we detected and subsequently addressed certain inaccuracies in Figure 2c. The corrected version was marked in red and Fig 2C was graphed. Though the value still seems to be high, the probable reason lying in the phenomenon may be the change of experimental conditions and original CFU the mixture contains. Additionally, the observed discrepancies may be attributed to differences in experimental conditions, particularly in the one-step growth curve protocol where bacterial cultures were resuspended in fresh medium. Furthermore, variations in the incubation time allocated for bacterial adsorption could also contribute to these differences. Similar observations have been reported in other studies³.

I. 138 What is the objective of testing the phage under different conditions? Unless some of the factors tested were likely found in physiological conditions or for maintaining the phage for a long time, this section seems unjustified.

Thank you for your detailed review. In response to your valuable suggestion, we have incorporated the phage stability analysis at various temperatures into the main text, while relocating the stability experiments involving UV light and organic reagents to the supplementary material for enhanced readability and focus.

I. Is Softberry software for gene annotation? Clarify.

Softberry serves as an online ORF prediction software. Instead, the functions of genes were predicted by BLASTp.

I. 154-158. In my opinion, it is unnecessary to report on the restriction sites unless the objective of the manuscript is genetic modification. Moreover, the restriction enzyme was not correctly addressed: lack of the undigested control, and fragments are not in the correct size.

Thank you for your suggestions! To enhance the functional annotation and biological relevance of our genomic analysis, we employed circular genome maps with comprehensive gene function annotations, replacing the previous version that primarily highlighted enzymatic sites. Furthermore, through comprehensive restriction enzyme digestion analysis, we experimentally

validated the circular conformation of the henu8 genome. We've also noticed the miss of "control". Thus, another electrophoresis with "control" was complemented to our article, and the fragments were carefully checked for credibility.

Fig. 6. It is unclear if the phage effects on "preformed biofilm" and "biofilm inhibition" are two different phenomena or the same. There are no distinctions between B and C plots (the letters are missing). It could be necessary to compare the columns side by side and obtain a statistic for the comparison. The term "aspirated" is not usual.

Thank you for your suggestions! We've added the missing mark to distinguish the "performed biofilm" and "biofilm inhibition". "Preformed biofilm" stands for the disruption ability of the biofilms already formed, where phage Henu8 was added to the wells 24h after the formation of bacterial biofilm, a shield is ready. On the contrary, "Biofilm inhibition" stands for the inhibition ability of the biofilm by co-treating the bacteria and phage Henu8 once the incubation began. The missing letters were added. As indicated, the inhibition test of Henu8 exerted a relatively strong anti-biofilm ability compared with that of the destruction test, though the differences are minor among different MOIs.

Fig. 7 What is the correlation between phage titers and biofilm formation and between phage titers and incubation time?

The lack of a linear correlation between phage titer and biofilm inhibition can be attributed to the repeated cycles of phage infection that occur during the incubation period with bacterial cells. In the growth curve analysis, no statistically significant differences were observed among the various phage titers. Furthermore, the one-step growth curve experiments revealed that bacterial populations rapidly developed phage resistance within a remarkably short timeframe, consequently eliminating any direct correlation between incubation duration and biofilm inhibition efficacy.

During our thorough analysis of the experimental data, we identified potential inconsistencies in the results presented in Figure 7. After careful consideration and rigorous verification, we have determined that these findings may not meet the stringent standards required for scientific publication. Consequently, we have decided to exclude this figure from the manuscript to ensure the overall scientific integrity and clarity of our work. This editorial decision will enable readers to better comprehend the core findings and maintain the highest standards of scientific accuracy in

our publication.

Fig. 8. How valid is the experiment if the Henu8 phage does not eliminate the cells? The authors show that after 4 hours of incubation, the remnant cells recover to growth. The experiment here is cut after three hours. Otherwise, explain why this incubation time was selected. I also see that this and the following experiments in mice are the most critical sections in the manuscript.

Thank you for the question. Most of the phages reported so far are susceptible to resistance in vitro but have good therapeutic effects in vivo⁴. The reason may be that in vitro the bacteria are larger in number and prone to resistant mutations. In vivo, immune cells are helpful in the treatment of phages. Bacteria that develop resistance to phages do so at the cost of adaptive mutations and may therefore be extremely sensitive to antibiotics⁵. At the same time, antibiotics also inhibit bacteria from developing resistance to phages⁶. Therefore, phages and antibiotics have a synergistic effect in the bactericidal process. In our experimental observations, no bacterial growth (mutation production) was detected within the initial 8-hour period. A modest and irregular increase in bacterial count was observed after this time frame. Consequently, this study focuses exclusively on investigating the synergistic effects of henu8 in combination with antibiotics. We appreciate your insightful question, which has prompted us to consider further exploration of bacterial responses under the dual selective pressure of henu8 and antibiotics in our future research.

Discussion.

It should be limited once the focus of the manuscript is defined.

Thank you for suggestion. The full text has been completely overhauled in response to reviewer and editorial comments.

- 1 Zhang, K. *et al.* Interactions between Viral Regulatory Proteins Ensure an MOI-Independent Probability of Lysogeny during Infection by Bacteriophage P1. *mBio* **12**, e0101321, doi:10.1128/mBio.01013-21 (2021).
- 2 Benala, M. *et al.* A revisited two-step microtiter plate assay: Optimization of in vitro multiplicity of infection (MOI) for Coliphage and Vibriophage. *J Virol Methods* **294**, 114177, doi:10.1016/j.jviromet.2021.114177 (2021).
- 3 Han, P. *et al.* Characterization of the Bacteriophage BUCT603 and Therapeutic Potential Evaluation Against Drug-Resistant *Stenotrophomonas maltophilia* in a Mouse Model. *Front Microbiol* **13**, 906961, doi:10.3389/fmicb.2022.906961 (2022).
- 4 Han, P., Pu, M., Li, Y., Fan, H. & Tong, Y. Characterization of bacteriophage BUCT631 lytic for K1 *Klebsiella pneumoniae* and its therapeutic efficacy in *Galleria mellonella* larvae. *Virus Sin* **38**, 801-812, doi:10.1016/j.virus.2023.07.002 (2023).
- 5 Cao, X. *et al.* Enhanced bacteriostatic effects of phage vB_C4 and cell wall-targeting antibiotic combinations against drug-resistant *Aeromonas veronii*. *Microbiology spectrum* **13**, e0190824, doi:10.1128/spectrum.01908-24 (2025).
- 6 Parab, L. *et al.* Chloramphenicol and gentamicin reduce the evolution of resistance to phage PhiX174 by suppressing a subset of *E. coli* LPS mutants. *PLoS biology* **23**, e3002952, doi:10.1371/journal.pbio.3002952 (2025).

Re: Spectrum01633-24R2 (**The virulent bacteriophage Henu8 as an antimicrobial synergist against *Escherichia coli***)

Dear Dr. qiming li:

Your manuscript has been accepted, and I am forwarding it to the ASM production staff for publication. Your paper will first be checked to make sure all elements meet the technical requirements. ASM staff will contact you if anything needs to be revised before copyediting and production can begin. Otherwise, you will be notified when your proofs are ready to be viewed.

Sincerely,
David Pride
Editor
Microbiology Spectrum

Reviewer #1 (Comments for the Author):

This has been well revised according to the comments.

Reviewer #3 (Comments for the Author):

The MOI question still remains: how low MOIs (0.01) have higher effect than MOIs of 1 or higher?